# Ensemble cryo-EM uncovers inchworm-like translocation of a viral IRES through the ribosome

**Priyanka D Abeyrathne[1†], Cha San Koh[2†], Timothy Grant[1†], Nikolaus Grigorieff[1]\*, Andrei A Korostelev[2]\***

[1]Janelia Research Campus, Howard Hughes Medical Institute, Ashburn, United States; [2]RNA Therapeutics Institute, Department of Biochemistry and Molecular Pharmacology, University of Massachusetts Medical School, Worcester, United States

**Abstract** Internal ribosome entry sites (IRESs) mediate cap-independent translation of viral mRNAs. Using electron cryo-microscopy of a single specimen, we present five ribosome structures formed with the Taura syndrome virus IRES and translocase eEF2•GTP bound with sordarin. The structures suggest a trajectory of IRES translocation, required for translation initiation, and provide an unprecedented view of eEF2 dynamics. The IRES rearranges from extended to bent to extended conformations. This inchworm-like movement is coupled with ribosomal inter-subunit rotation and 40S head swivel. eEF2, attached to the 60S subunit, slides along the rotating 40S subunit to enter the A site. Its diphthamide-bearing tip at domain IV separates the tRNA-mRNA-like pseudoknot I (PKI) of the IRES from the decoding center. This unlocks 40S domains, facilitating head swivel and biasing IRES translocation *via* hitherto-elusive intermediates with PKI captured between the A and P sites. The structures suggest missing links in our understanding of tRNA translocation.

**\*For correspondence:** niko@grigorieff.org (NG); andrei.korostelev@umassmed.edu (AAK)

[†]These authors contributed equally to this work

## Introduction

Virus propagation relies on the host translational apparatus. To efficiently compete with host mRNAs and engage in translation under stress, some viral mRNAs undergo cap-independent translation. To this end, internal ribosome entry site (IRES) RNAs are employed (reviewed in *Deforges et al. (2015)*, *Jackson et al. (2010)*, *Lozano and Martinez-Salas (2015)*. An IRES is located at the 5' untranslated region of the viral mRNA, preceding an open reading frame (ORF). To initiate translation, a structured IRES RNA interacts with the 40S subunit or the 80S ribosome, resulting in precise positioning of the downstream start codon in the small 40S subunit. The canonical scenario of cap-dependent and IRES-dependent initiation involves positioning of the AUG start codon and the initiator tRNA[Met] in the ribosomal *p*eptidyl-tRNA (P) site, facilitated by interaction with initiation factors (*Jackson et al., 2010*). Subsequent binding of an elongator *a*minoacyl-tRNA to the ribosomal A site transitions the initiation complex into the elongation cycle of translation. Upon peptide bond formation, the two tRNAs and their respective mRNA codons translocate from the A and P to P and E (exit) sites, freeing the A site for the next elongator tRNA.

An unusual strategy of initiation is used by intergenic-region (IGR) IRESs found in *Dicistroviridae* arthropod-infecting viruses. These include shrimp-infecting Taura syndrome virus (TSV; *Cevallos and Sarnow, 2005*; *Hatakeyama et al., 2004*), and insect viruses Plautia stali intestine virus (PSIV; *Nishiyama et al. (2003)*, *Sasaki and Nakashima (1999)*) and Cricket paralysis virus (CrPV; *Jan et al. (2001)*, *Wilson et al. (2000)*). The IGR IRES mRNAs do not contain an AUG start codon. The IGR-IRES-driven initiation does not involve initiator tRNA[Met] and initiation factors (*Jan et al., 2001*;

*Sasaki and Nakashima, 1999*; *Wilson et al., 2000*). As such, this group of IRESs represents the most streamlined mechanism of eukaryotic translation initiation. A recent demonstration of bacterial translation initiation by an IGR IRES (*Colussi et al., 2015*) indicates that the IRESs take advantage of conserved structural and dynamic properties of the ribosome. Early electron cryo-microscopy (cryo-EM) studies have found that the CrPV IRES packs in the ribosome intersubunit space (*Schuler et al., 2006*; *Spahn et al., 2004b*). Recent cryo-EM structures of ribosome-bound TSV IRES (*Koh et al., 2014*) and CrPV IRES (*Fernandez et al., 2014*) revealed that IGR IRESs position the ORF by mimicking a translating ribosome bound with tRNA and mRNA. The ~200-nt IRES RNAs span from the A site beyond the E site. A conserved tRNA-mRNA–like structural element of pseudoknot I (PKI; *Costantino et al., 2008*) interacts with the decoding center in the A site of the 40S subunit (*Fernandez et al., 2014*; *Koh et al., 2014*). The codon-anticodon-like helix of PKI is stabilized by interactions with the universally conserved decoding-center nucleotides G577, A1755 and A1756 (G530, A1492 and A1493 in *E. coli* 16S ribosomal RNA, or rRNA). The downstream initiation codon—coding for alanine—is placed in the mRNA tunnel, preceding the decoding center. PKI of IGR IRESs therefore mimics an A-site elongator tRNA interacting with an mRNA sense codon, but not a P-site initiator tRNA^Met and the AUG start codon.

How this non-canonical initiation complex transitions to the elongation step is not fully understood. For a cognate aminoacyl-tRNA to bind the first viral mRNA codon, PKI has to be translocated from the A site, so that the first codon can be presented in the A site. A cryo-EM structure of the ribosome bound with a CrPV IRES and release factor eRF1 occupying the A site provided insight into the post-translocation state (*Muhs et al., 2015*). In this structure, PKI is positioned in the P site and the first mRNA codon is located in the A site. How the large IRES RNA translocates within the ribosome, allowing PKI translocation from the A to P site is not known.

The structural similarity of PKI and the tRNA anticodon stem loop (ASL) bound to a codon suggests that their mechanisms of translocation are similar to some extent. Translocation of the IRES or tRNA-mRNA requires eukaryotic elongation factor 2 (eEF2) (*Jan et al., 2003*; *Pestova and Hellen, 2003*), a structural and functional homolog of the well-studied bacterial EF-G (*Czworkowski et al., 1994*; *Evarsson et al., 1994*). Pre-translocation tRNA-bound ribosomes contain a peptidyl- and deacyl-tRNA, both base-paired to mRNA codons in the A and P sites (termed 2tRNA•mRNA complex). Translocation of 2tRNA•mRNA involves two major large-scale ribosome rearrangements (*Figure 1—figure supplement 1*) (reviewed in *Ling and Ermolenko, (2016)*). First, studies of bacterial ribosomes showed that a ~10° rotation of the small subunit relative to the large subunit, known as intersubunit rotation, or ratcheting (*Frank and Agrawal, 2000*), is required for translocation (*Horan and Noller, 2007*). Intersubunit rotation occurs spontaneously upon peptidyl transfer, and is coupled with formation of hybrid tRNA states (*Agirrezabala et al., 2008*; *Blanchard et al., 2004*; *Cornish et al., 2008*; *Ermolenko et al., 2007*; *Julián et al., 2008*; *Moazed and Noller, 1989*). In the rotated pre-translocation ribosome, the peptidyl-tRNA binds the A site of the small subunit with its ASL and the P site of the large subunit with the CCA 3' end (A/P hybrid state). Concurrently, the deacyl-tRNA interacts with the P site of the small subunit and the E site of the large subunit (P/E hybrid state). The ribosome can undergo spontaneous, thermally-driven forward-reverse rotation (*Cornish et al., 2008*) that shifts the two tRNAs between the hybrid and 'classical' states while the anticodon stem loops remain non-translocated. Binding of EF-G next to the A site and reverse rotation of the small subunit results in translocation of both ASLs on the small subunit (*Ermolenko and Noller, 2011*). EF-G is thought to 'unlock' the pre-translocation ribosome (*Savelsbergh et al., 2003*; *Spirin, 1969*), allowing movement of the 2tRNA•mRNA complex, however the structural details of this unlocking are not known.

The second large-scale rearrangement involves rotation, or swiveling, of the head of the small subunit relative to the body. The head can rotate by up to ~20° around the axis nearly orthogonal to that of intersubunit rotation, in the absence of tRNA (*Schuwirth et al., 2005*) or in the presence of a single P/E tRNA and eEF2 (*Taylor et al., 2007*) or EF-G (*Ratje et al., 2010*). Förster resonance energy transfer (FRET) data suggest that head swivel of the rotated small subunit facilitates EF-G-mediated movement of 2tRNA•mRNA (*Guo and Noller, 2012*). Structures of the 70S•EF-G complex bound with two nearly translocated tRNAs (*Ramrath et al., 2013*; *Zhou et al., 2014*), exhibit a large 18° to 21° head swivel in a mid-rotated subunit, whereas no head swivel is observed in the fully rotated pre-translocation or in the non-rotated post-translocation 70S•2tRNA•EF-G structures (*Brilot et al., 2013*; *Gao et al., 2009*). The structural role of head swivel is not fully understood. The

head swivel was proposed to facilitate transition of the tRNA from the P to E site by widening a constriction between these sites on the 30S subunit (*Schuwirth et al., 2005*). This widening allows the ASL to sample positions between the P and E sites (*Ratje et al., 2010*). Whether and how the head swivel mediates tRNA transition from the A to P site remains unknown.

We sought to address the following questions by structural visualization of 80S•IRES•eEF2 translocation complexes: (1) How does a large IRES RNA move through the restricted intersubunit space, bringing PKI from the A to P site of the small subunit? (2) How does eEF2 mediate IRES translocation? (3) Does IRES translocation involve large rearrangements in the ribosome, similar to tRNA translocation? (4) What, if any, is the mechanistic role of 40S head rotation in IRES translocation? We used cryo-EM to visualize 80S•TSV IRES complexes formed in the presence of eEF2•GTP and the translation inhibitor sordarin, which stabilizes eEF2 on the ribosome. Although the mechanism of sordarin action is not fully understood, the inhibitor does not affect the conformation of eEF2•GDPNP on the ribosome (*Taylor et al., 2007*), rendering it an excellent tool in translocation studies. Maximum-likelihood classification using FREALIGN (*Lyumkis et al., 2013*) identified five IRES-eEF2-bound ribosome structures within a single sample (*Figures 1* and *2*). The structures differ in the positions and conformations of ribosomal subunits (*Figures 1b* and *2*), IRES RNA (*Figures 3* and *4*) and eEF2 (*Figures 5* and *6*). This ensemble of structures allowed us to reconstruct a sequence of steps in IRES translocation induced by eEF2.

## Results

We used single-particle cryo-EM and maximum-likelihood image classification in FREALIGN to obtain three-dimensional density maps from a single specimen. The translocation complex was formed using *S. cerevisiae* 80S ribosomes, Taura syndrome virus IRES, and *S. cerevisiae* eEF2 in the presence of GTP and the eEF2-binding translation inhibitor sordarin. Unsupervised cryo-EM data classification was combined with the use of three-dimensional and two-dimensional masking around the ribosomal A site (*Figure 1—figure supplement 2*). This approach revealed five 80S•IRES•eEF2•GDP structures at average resolutions of 3.5 to 4.2 Å, sufficient to locate IRES domains and to resolve individual residues in the core regions of the ribosome and eEF2 (*Figures 3c,d,* and *5f,h*; see also *Figure 1—figure supplement 2* and *Figure 5—figure supplement 2*), including the post-translational modification diphthamide 699 (*Figure 3c*).

Our structures represent hitherto uncharacterized translocation complexes of the TSV IRES captured within globally distinct 80S conformations (*Figures 1b* and *2*). We numbered the structures from I to V, according to the position of the tRNA-mRNA-like PKI on the 40S subunit (*Figure 2—source data 1*). Specifically, PKI is partially withdrawn from the A site in Structure I, and fully translocated to the P site in Structure V (*Figure 4*; see also *Figure 3—figure supplement 1*). Thus Structures I to IV represent different positions of PKI between the A and P sites (*Figure 2—source data 1*), suggesting that these structures describe intermediate states of translocation. Structure V corresponds to the post-translocation state.

### Changes in ribosome conformation and eEF2 positions are coupled with IRES movement through the ribosome

#### Intersubunit rotation

Using the post-translocation *S. cerevisiae* 80S ribosome bound with the P and E site tRNAs as a reference (80S•2tRNA•mRNA), in which both the subunit rotation and the head-body swivel are 0° (*Svidritskiy et al., 2014*), we found that the ribosome adopts four globally distinct conformations in Structures I through V (*Figure 1b*; see also *Figure 1—figure supplement 1* and *Figure 2—source data 1*). Structure I comprises the most rotated ribosome conformation (~10°), characteristic of pretranslocation hybrid-tRNA states. From Structure I to V, the body of the small subunit undergoes backward (reverse) rotation (*Figure 2b*; see also *Figure 1—figure supplement 2* and *Figure 2—figure supplement 1*). Structures II and III are in mid-rotation conformations (~5°). Structure IV adopts a slightly rotated conformation (~1°). Structure V is in a nearly non-rotated conformation (0.5°), very similar to that of post-translocation ribosome-tRNA complexes (*Gao et al., 2009*; *Korostelev et al., 2006*; *Selmer et al., 2006*; *Svidritskiy et al., 2014*). Thus, intersubunit rotation of ~9° from Structure I to V covers a nearly complete range of relative subunit positions, similar to what was reported

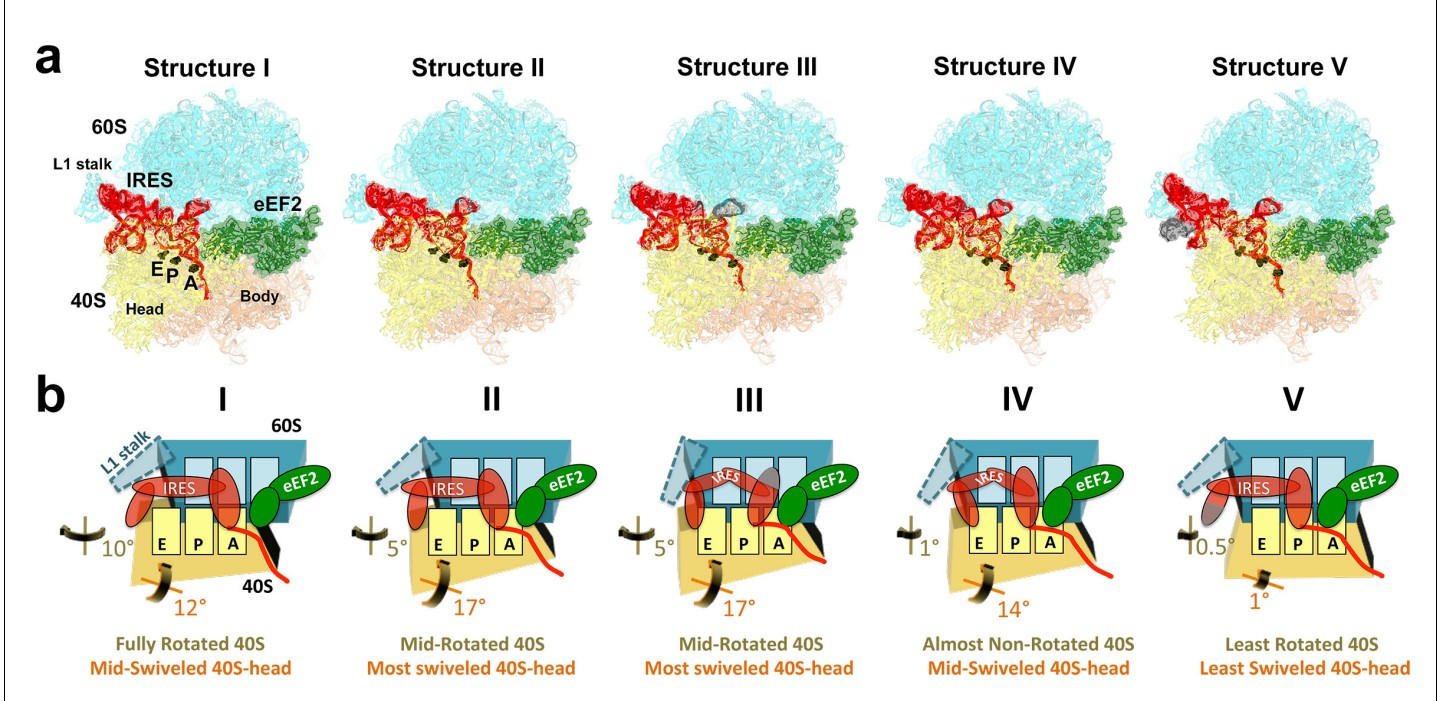

**Figure 1.** Cryo-EM structures of the 80S•TSV IRES bound with eEF2•GDP•sordarin. (a) Structures I through V. In all panels, the large ribosomal subunit is shown in cyan; the small subunit in light yellow (head) and wheat-yellow (body); the TSV IRES in red, eEF2 in green. Nucleotides C1274, U1191 of the 40S head and G904 of the platform (C1054, G966 and G693 in *E. coli* 16S rRNA) are shown in black to denote the A, P and E sites, respectively. Unresolved regions of the IRES in densities for Structures III and V are shown in gray. (b) Schematic representation of the structures shown in panel a, denoting the conformations of the small subunit relative to the large subunit. A, P and E sites are shown as rectangles. All measurements are relative to the non-rotated 80S•2tRNA•mRNA structure (*Svidritskiy et al., 2014*). The colors are as in panel a.

The following source data and figure supplements are available for figure 1:

**Source data 1.** Structure refinement statistics for Structures I, II, III, IV, V.

**Figure supplement 1.** Comparison of 70S•2tRNA•mRNA and 80S•IRES translocation complexes.

**Figure supplement 2.** Schematic of cryo-EM refinement and classification procedures.

**Figure supplement 3.** Cryo-EM density of Structures I-V.

for tRNA-bound yeast (*Svidritskiy et al., 2014*; *Taylor et al., 2007*), bacterial (*Agirrezabala et al., 2008*; *Fischer et al., 2010*; *Julián et al., 2008*) and mammalian (*Budkevich et al., 2011*) ribosomes.

## 40S head swivel

The pattern of 40S head swivel between the structures is different from that of intersubunit rotation (*Figures 2c and d*; see also *Figure 2—source data 1*). As with the intersubunit rotation, the small head swivel (~1°) in the non-rotated Structure V is closest to that in the 80S•2tRNA•mRNA post-translocation ribosome (*Svidritskiy et al., 2014*). However in the pre-translocation intermediates (from Structure I to IV), the beak of the head domain first turns toward the large subunit and then backs off (*Figure 2—figure supplement 1*). This movement reflects the forward and reverse swivel. The head samples a mid-swiveled position in Structure I (12°), then a highly-swiveled position in Structures II and III (17°) and a less swiveled position in Structure IV (14°). The maximum head swivel is observed in the mid-rotated complexes II and III, in which PKI transitions from the A to P site, while eEF2 occupies the A site partially. By comparison, the similarly mid-rotated (4°) 80S•TSV IRES initiation complex, in the absence of eEF2 (*Koh et al., 2014*), adopts a mid-swiveled position (~10°)

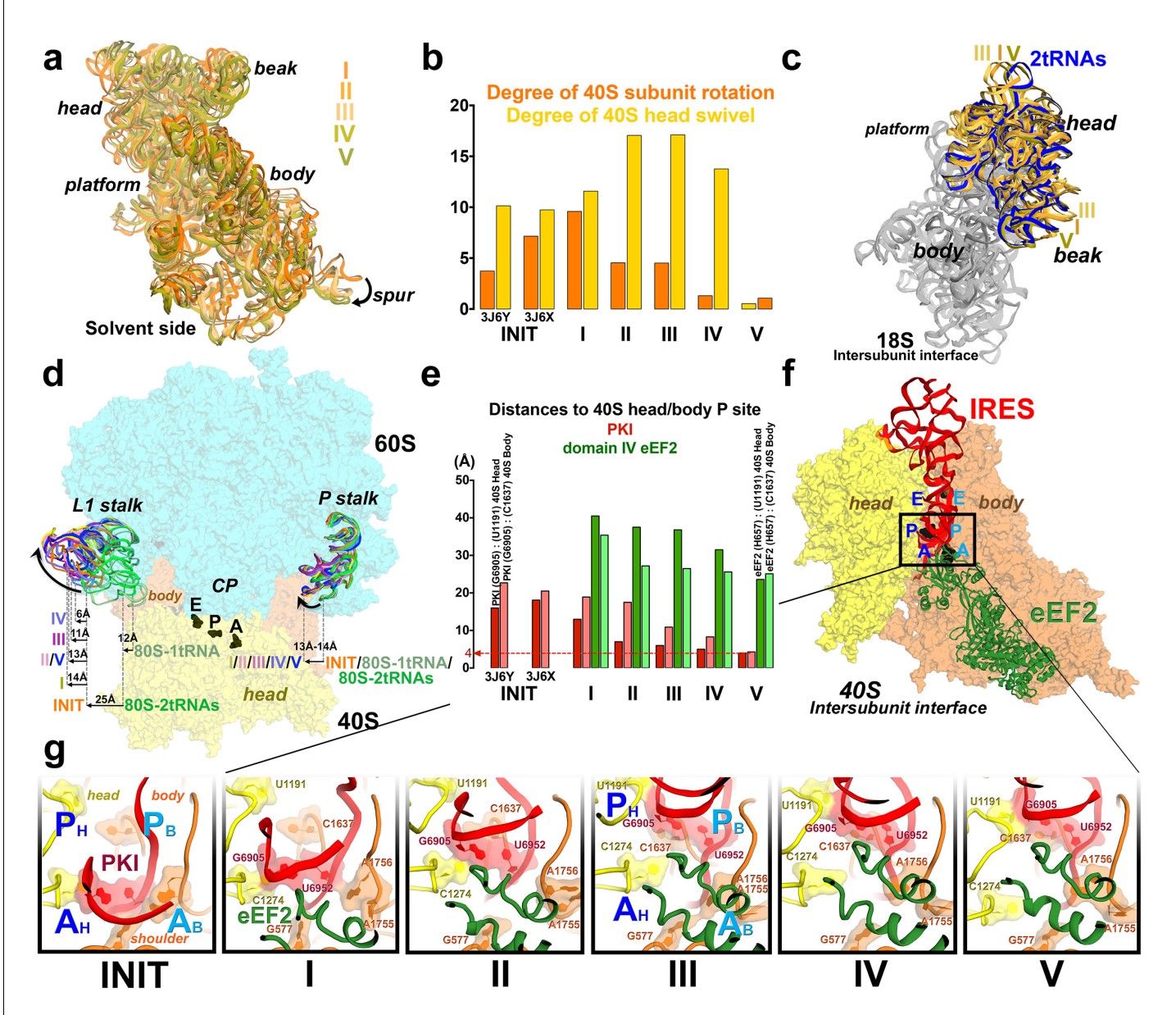

**Figure 2.** Large-scale rearrangements in Structures I through V, coupled with the movement of PKI from the A to P site and eEF2 entry into the A site. (a) Comparison of the 40S-subunit rotational states in Structures I through V, sampling a ~10° range between Structure I (fully rotated) and Structure V (non-rotated). 18S ribosomal RNA is shown and ribosomal proteins are omitted for clarity. The superpositions of Structures I-V were performed by structural alignments of the 25S ribosomal RNAs. (b) Bar graph of the angles characterizing the 40S rotational and 40S head swiveling states in Structures I through V. Measurements for the two 80S•IRES (INIT) structures (**Koh et al., 2014**) are included for comparison. All measurements are relative to the non-rotated 80S•2tRNA•mRNA structure (**Svidritskiy et al., 2014**). (c) Comparison of the 40S conformations in Structures I through V shows distinct positions of the head relative to the body of the 40S subunit (head swivel). Conformation of the non-swiveled 40S subunit in the *S. cerevisiae* 80S ribosome bound with two tRNAs (**Svidritskiy et al., 2014**) is shown for reference (blue). (d) Comparison of conformations of the L1 and P stalks of the large subunit in Structures I through V with those in the 80S•IRES (**Koh et al., 2014**) and tRNA-bound 80S (**Svidritskiy et al., 2014**) structures. Superpositions were performed by structural alignments of 25S ribosomal RNAs. The central protuberance (CP) is labeled. (e) Bar graph of the positions of PKI and domain IV of eEF2 relative to the P site residues of the head (U1191) and body (C1637) in Structures I through V. (f and g) Close-up view of rearrangements in the A and P sites from the initiation state (INIT: PDB ID 3J6Y) to the post-translocation Structure V. The fragment shown within a rectangle in panel f is magnified in panel g. Nucleotides of the 40S body are shown in orange, 40S head in yellow. The superpositions of structures were performed by structural alignments of the 18S ribosomal RNAs excluding the head region (nt 1150–1620).

The following source data and figure supplement are available for figure 2:

*Figure 2 continued on next page*

Figure 2 continued

**Source data 1.** Measurements for conformations and positions in Structures I through V.

**Figure supplement 1.** Large-scale rearrangements in Structures I through V, coupled with the movement of PKI from the A to P site and eEF2 entry into the A site.

(*Figure 2c*). These observations suggest that eEF2 is necessary for inducing or stabilizing the large head swivel of the 40S subunit characteristic for IRES translocation intermediates.

## IRES rearrangements

In each structure, the TSV IRES adopts a distinct conformation in the intersubunit space of the ribosome (*Figures 3* and *4*). The IRES (nt 6758–6952) consists of two globular parts (*Figure 3a*): the 5′-region (domains I and II, nt 6758–6888) and the PKI domain (domain III, nt 6889–6952). We collectively term domains I and II the 5′ domain. The PKI domain comprises PKI and stem loop 3 (SL3), which stacks on top of the stem of PKI (*Koh et al., 2014*). The $^{6953}$GCU triplet immediately following the PKI domain is the first codon of the open reading frame. In the eEF2-free 80S•IRES initiation complex (INIT) (*Koh et al., 2014*), the bulk of the 5′-domain (nt. 6758–6888) binds near the E site, contacting the ribosome mostly by means of three protruding structural elements: the L1.1 region and stem loops 4 and 5 (SL4 and SL5). In Structures I to IV, these contacts remain as in the initiation complex (*Figure 1a*). Specifically, the L1.1 region interacts with the L1 stalk of the large subunit, while SL4 and SL5 bind at the side of the 40S head and interact with proteins uS7, uS11 and eS25 (*Figure 3—figure supplement 2* and *Figure 3—figure supplement 3*; ribosomal proteins are termed according to *Ban et al., 2014*). In Structures I-IV, the minor groove of SL4 (at nt 6840–6846) binds next to an α-helix of uS7, which is rich in positively charged residues (K212, K213, R219 and K222). The tip of SL4 binds in the vicinity of R157 in the β-hairpin of uS7 and of Y58 in uS11. The minor groove of SL5 (at nt 6862–6868) contacts the positively charged region of eS25 (R49, R58 and R68) (*Figure 3—figure supplement 4*). In Structure V, however, the density for SL5 is missing suggesting that SL5 is mobile, while weak SL4 density suggests that SL4 is shifted along the surface of uS7, ∼20 Å away from its initial position (*Figure 3—figure supplement 2c*). The L1.1 region remains in contact with the L1 stalk (*Figure 3—figure supplement 3*).

The shape of the IRES changes considerably from the initiation state to Structures I through V, from an extended to compact to extended conformation (*Figure 4*; see also *Figure 3—figure supplement 2a*). Because in Structures I to IV the PKI domain shifts toward the P site, while the 5′ remains unchanged near the E site, the distance between the domains shortens (*Figure 4*). In the 80S•IRES initiation state (*Koh et al., 2014*), the A-site-bound PKI is separated from SL4 by almost 50 Å (*Figure 4*). In Structures I and II, the PKI is partially retracted from the A site and the distance from SL4 shortens to ∼35 Å. As PKI moves toward the P site in Structures III and IV, the PKI domain approaches to within ∼25 Å of SL4. Because the 5′-domain in the following structure (V) moves by ∼20 Å along the 40S head, the IRES returns to an extended conformation (∼45 Å) that is similar to that in the 80S•IRES initiation complex.

Rearrangements of the IRES involve restructuring of several interactions with the ribosome. In Structure I, SL3 of the PKI domain is positioned between the A-site finger (nt 1008–1043 of 25S rRNA) and the P site of the 60S subunit, comprising helix 84 of 25S rRNA (nt. 2668–2687) and protein uL5 (*Figure 3—figure supplement 6*). This position of SL3 is ∼25 Å away from that in the 80S•IRES initiation state (*Koh et al., 2014*), in which PKI and SL3 closely mimic the ASL and elbow of the A-site tRNA, respectively (*Koh et al., 2014*). As such, the transition from the initiation state to Structure I involves repositioning of SL3 around the A-site finger, resembling the transition between the pre-translocation A/P and A/P* tRNA (*Brilot et al., 2013*). The second set of major structural changes involves interaction of the P site region of the large subunit with the hinge point of the IRES bending between the 5′ domain and the PKI domain (nt. 6886–6890). In the highly bent Structures III and IV, the hinge region interacts with the universally conserved uL5 and the C-terminal tail of eL42 (*Figure 3—figure supplement 7*). However, in the extended conformations, these parts of the IRES and the 60S subunit are separated by more than 10 Å, suggesting that an interaction between them stabilizes the bent conformations but not the extended ones. Another local rearrangement concerns

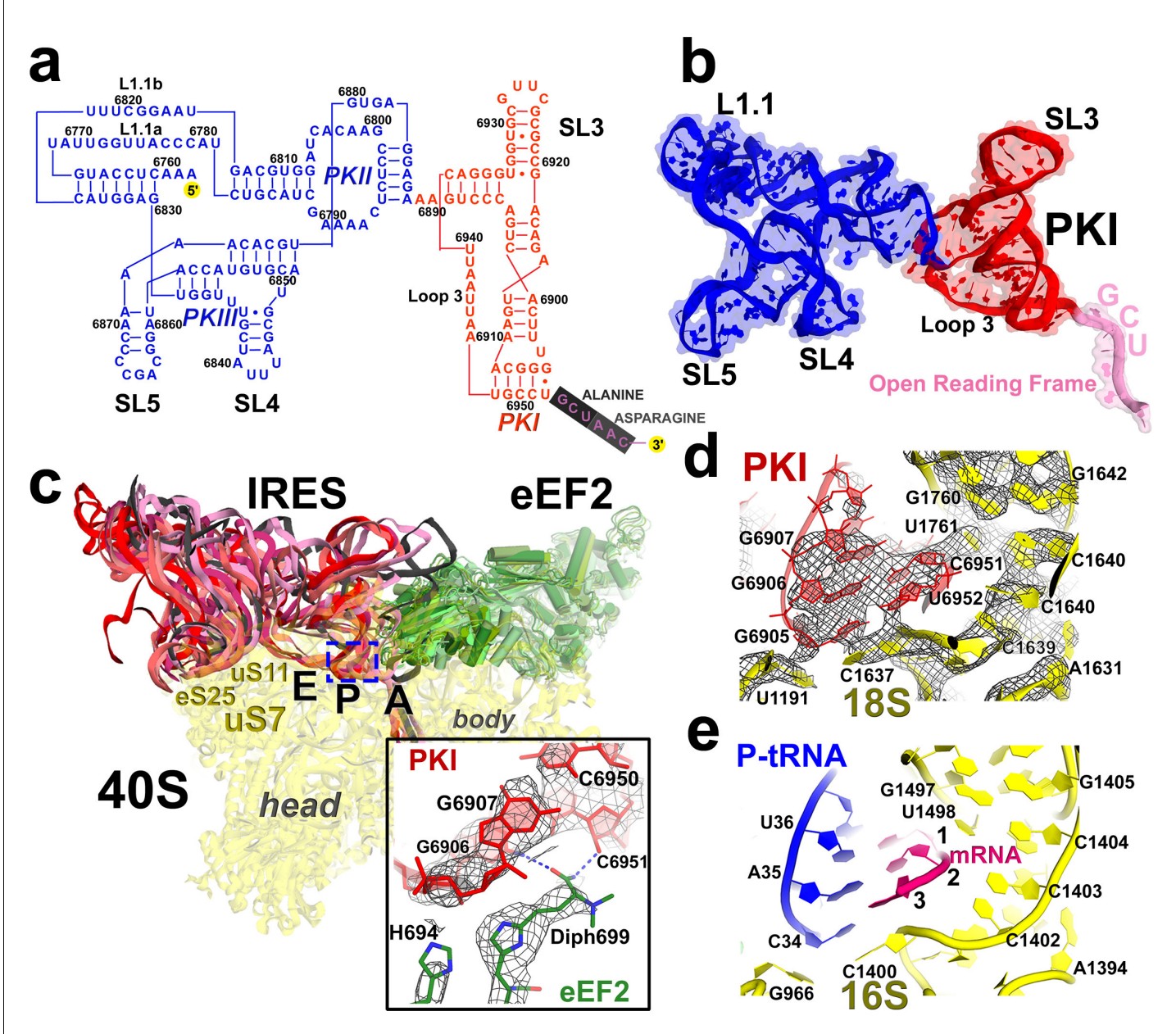

**Figure 3.** Positions of the IRES relative to eEF2 and elements of the ribosome in Structures I through V. (a) Secondary structure of the TSV IRES. The TSV IRES comprises two domains: the 5′ domain (blue) and the PKI domain (red). The open reading frame (gray) is immediately following pseudoknot I (PKI). (b) Three-dimensional structure of the TSV IRES (Structure II). Pseudoknots and stem loops are labeled and colored as in (a). (c) Positions of the IRES and eEF2 on the small subunit in Structures I to V. The initiation-state IRES is shown in gray. The insert shows density for interaction of diphthamide 699 (eEF2; green) with the codon-anticodon-like helix (PKI; red) in Structure V. (d and e) Density of the P site in Structure V shows that interactions of PKI with the 18S rRNA nucleotides (c) are nearly identical to those in the P site of the 2tRNA•mRNA-bound 70S ribosome (d; *Svidritskiy et al., 2013*).

The following figure supplements are available for figure 3:

**Figure supplement 1.** Comparison of the TSV IRES and eEF2 positions in Structures I through V.

**Figure supplement 2.** Positions of the IRES relative to proteins uS7, uS11 and eS25.

**Figure supplement 3.** Positions of the L1stalk, tRNA and TSV IRES relative to proteins uS7 and eS25, in 80S•tRNA structures (*Svidritskiy et al., 2014*) and 80S•IRES structures I and V (this work).

*Figure 3 continued on next page*

*Figure 3 continued*

**Figure supplement 4.** Interactions of the stem loops 4 and 5 of the TSV with proteins uS7 and eS25.

**Figure supplement 5.** Position and interactions of loop 3 (variable loop region) of the PKI domain in Structure V (this work) resembles those of the anticodon stem loop of the E-site tRNA (blue) in the 80S•2tRNA•mRNA complex (*Svidritskiy et al., 2014*).

**Figure supplement 6.** Positions of tRNAs and the TSV IRES relative to the A-site finger (blue, nt 1008–1043 of 25S rRNA) and the P site of the large subunit, comprising helix 84 of 25S rRNA (nt. 2668–2687) and protein uL5 (collectively labeled as central protuberance, CP, in the upper-row first figure, and individually labeled in the lower-row first figure).

**Figure supplement 7.** Interactions of the TSV IRES with uL5 and eL42.

loop 3, also known as the variable loop region (*Ruehle et al., 2015*; *Ren et al., 2014*; *Au and Jan, 2012*), which connects the ASL- and mRNA-like parts of PKI. This loop is poorly resolved in Structures I through IV, suggesting conformational flexibility in agreement with structural studies of the isolated PKI (*Costantino et al., 2008*; *Zhu et al., 2011*) and biochemical studies of unbound IRESs (*Jan and Sarnow, 2002*; *Pfingsten et al., 2010*). In Structure V, loop 3 is bound in the 40S E site and the backbone of loop 3 near the codon-like part of PKI (at nt. 6945–6946) interacts with R148 and R157 in β-hairpin of uS7. The interaction of loop 3 backbone with uS7 resembles that of the anticodon-stem loop of E-site tRNA in the post-translation 80S•2tRNA•mRNA structure (*Figure 3—figure supplement 5*) (*Svidritskiy et al., 2014*). Ordering of loop 3 suggests that this flexible region contributes to the stabilization of the PKI domain in the post-translocation state. This interpretation is consistent with the recent observation that alterations in loop 3 of the CrPV IRES result in decreased efficiency of translocation (*Ruehle et al., 2015*).

## eEF2 structures

Elongation factor eEF2 in all five structures is bound with GDP and sordarin (*Figure 5*). The elongation factor consists of three dynamic superdomains (*Jorgensen et al., 2003*): an N-terminal globular superdomain formed by the G (GTPase) domain (domain I) and domain II; a linker domain III; and a C-terminal superdomain comprising domains IV and V (*Figure 5a*). Domain IV extends from the main body and is critical for translocation catalyzed by eEF2 or EF-G. ADP-ribosylation of eEF2 at the tip of domain IV (*Davydova and Ovchinnikov, 1990*; *Nygard and Nilsson, 1990*) or deletion of domain IV from EF-G (*Martemyanov and Gudkov, 1999*; *Rodnina et al., 1997*) abrogate translocation. In post-translocation-like 80S•tRNA•eEF2 complexes, domain IV binds in the 40S A site, suggesting direct involvement of domain IV in translocation of tRNA from the A to P site (*Spahn et al., 2004a*; *Taylor et al., 2007*). GDP in our structures is bound in the GTPase center (*Figures 5d, e and f*) and sordarin is sandwiched between the β-platforms of domains III and V (*Figures 5g and h*), as in the structure of free eEF2•sordarin complex (*Jorgensen et al., 2003*).

The global conformations of eEF2 (*Figure 5a*) are similar in these structures (all-atom RMSD $\leq$ 2 Å), but the positions of eEF2 relative to the 40S subunit differ substantially as a result of 40S subunit rotation (*Figure 2—source data 1*). From Structure I to V, eEF2 is rigidly attached to the GTPase-associated center of the 60S subunit. The GTPase-associated center comprises the P stalk (L11 and L7/L12 stalk in bacteria) and the sarcin-ricin loop (SRL, nt 3012–3042). The tips of 25S rRNA helices 43 and 44 of the P stalk (nucleotides G1242 and A1270, respectively) stack on V754 and Y744 of domain V. An αββ motif of the eukaryote-specific protein P0 (aa 126–154) packs in the crevice between the long α-helix D (aa 172–188) of the GTPase domain and the β-sheet region (aa 246–263) of the GTPase domain insert (or G' insert) of eEF2 (secondary-structure nomenclatures for eEF2 and EF-G (*Czworkowski et al., 1994*) are the same). Although the P/L11 stalk is known to be dynamic (*Korostelev et al., 2008*; *Taylor et al., 2012*), its position remains unchanged from Structure I to V: all-atom root-mean-square differences for the 25S rRNA of the P stalk (nt 1223–1286) are within 2.5 Å. However, with respect to its position in the 80S•IRES complex in the absence of eEF2 and in the 80S•2tRNA•mRNA complex, the P stalk is shifted by ~13 Å toward the A site (*Figure 2d*). The sarcin-ricin loop interacts with the GTP-binding site of eEF2 (*Figures 5d and f*). While the overall

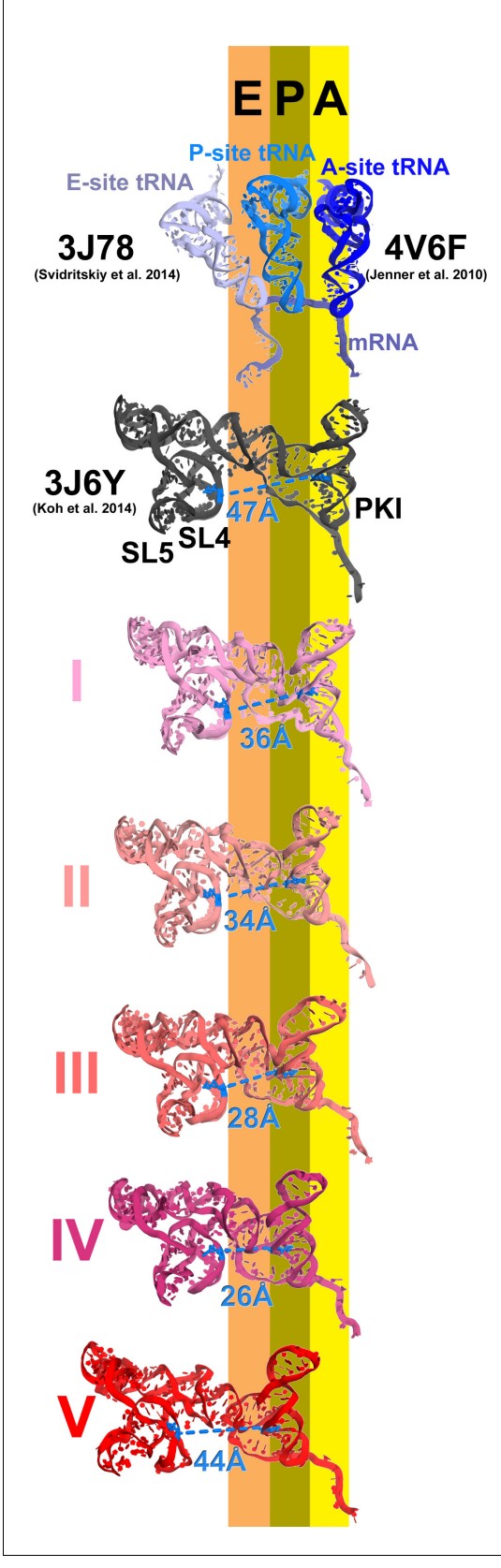

**Figure 4.** Inchworm-like translocation of the TSV IRES. Conformations and positions of the IRES in the

*Figure 4 continued on next page*

mode of this interaction is similar to that seen in 70S•EF-G crystal structures (*Chen et al., 2013b*; *Gao et al., 2009*; *Pulk and Cate, 2013*; *Tourigny et al., 2013*; *Zhou et al., 2013*; *2014*), there is an important local difference between Structure I and Structures II-V in switch loop I, as discussed below.

There are two modest but noticeable domain rearrangements between Structures I and V. Unlike in free eEF2, which can sample large movements of at least 50 Å of the C-terminal superdomain relative to the N-terminal superdomain (*Figure 5c*) (*Jorgensen et al., 2003*), eEF2 undergoes moderate repositioning of domain IV (~3 Å; *Figure 5a*) and domain III (~5 Å; *Figure 6d*). This limited flexibility of the ribosome-bound eEF2 is likely the result of simultaneous fixation of eEF2 superdomains, *via* domains I and V, by the GTPase-associated center of the large subunit. Domain IV of eEF2 binds at the 40S A site in Structures I to V but the mode of interaction differs in each complex (*Figure 6*). Because eEF2 is rigidly attached to the 60S subunit and does not undergo large inter-subunit rearrangements, gradual entry of domain IV into the A site between Structures I and V is due to 40S subunit rotation and head swivel. eEF2 settles into the A site from Structure I to V, as the tip of domain IV shifts by ~10 Å relative to the body and by ~20 Å relative to the swiveling head. Modest intra-eEF2 shifts of domain IV between Structures I to V outline a stochastic trajectory (*Figure 5a*), consistent with local adjustments of the domain in the A site. At the central region of eEF2, domains II and III contact the 40S body (mainly at nucleotides 48–52 and 429–432 of 18S rRNA helix 5 and uS12). From Structure I to V, these central domains migrate by ~10 Å along the 40S surface (*Figure 6c*). Comparison of eEF2 conformations reveals that in Structure V, domain III is displaced as a result of interaction with uS12, as discussed below.

In summary, between Structures I and V, a step-wise translocation of PKI by ~15 Å from the A to P site - within the 40S subunit – occurs simultaneously with the ~11 Å sideway entry of domain IV into the A site coupled with ~3 to 5 Å inter-domain rearrangements in eEF2. These shifts occur during the reverse rotation of the 40S body coupled with the forward-then-reverse head swivel. To elucidate the detailed structural mechanism of IRES translocation and the roles of eEF2 and ribosome rearrangements, we describe in the

*Figure 4 continued*

initiation state and in Structures I-V are shown relative to those of the A-, P- and E-site tRNAs. The view was obtained by structural alignment of the body domains of 18S rRNAs of the corresponding 80S structures. Distances between nucleotides 6848 and 6913 in SL4 and PKI, respectively, are shown (see also *Figure 2— source data 1*).

following sections the interactions of PKI and eEF2 with the ribosomal A and P sites in Structures I through V (*Figure 2g*; see also *Figure 1—figure supplement 1*).

## Structure I represents a pre-translocation IRES and initial entry of eEF2 in a GTP-like state

In the fully rotated Structure I, PKI is shifted toward the P site by ~3 Å relative to its position in the initiation complex but maintains interactions with the partially swiveled head. At the head, C1274 of the 18S rRNA (C1054 in *E. coli*) base pairs with the first nucleotide of the ORF immediately downstream of PKI. The C1274:G6953 base pair provides a stacking platform for the codon-anticodon–like helix of PKI. We therefore define C1274 as the foundation of the 'head A site'. Accordingly, we use U1191 (G966 in *E. coli*) and C1637 (C1400 in *E. coli*) as the reference points of the 'head P site' and 'body P site' (*Figure 2g*), respectively, because these nucleotides form a stacking foundation for the fully translocated mRNA-tRNA helix in tRNA-bound structures (*Korostelev et al., 2006*; *Selmer et al., 2006*; *Svidritskiy et al., 2014*) and in our post-translocation Structure V discussed below.

The interaction of PKI with the 40S body is substantially rearranged relative to that in the initiation state. In the latter, PKI is stabilized by interactions with the universally conserved decoding-center nucleotides G577, A1755 and A1756 ('body A site'), as in the A-site tRNA bound complexes (*Koh et al., 2014*). In Structure I, PKI does not contact these nucleotides (*Figures 2g* and *7*).

The position of eEF2 on the 40S subunit of Structure I is markedly distinct from those in Structures II to V. The translocase interacts with the 40S body but does not contact the head (*Figures 5b* and *6a*; *Figure 5—figure supplement 1*). Domain IV is partially engaged with the body A site. The tip of domain IV is wedged between PKI and decoding-center nucleotides A1755 and A1756, which are bulged out of h44. This tip contains the histidine-diphthamide triad (H583, H694 and Diph699), which interacts with the codon-anticodon-like helix of PKI and A1756 (*Figure 7*). Histidines 583 and 694 interact with the phosphate backbone of the anticodon-like strand (at G6907 and C6908). Diphthamide is a unique posttranslational modification conserved in archaeal and eukaryotic EF2 (at residue 699 in *S. cerevisiae*) and involves addition of a ~7-Å long 3-carboxyamido-3-(trimethylamino)-propyl moiety to the histidine imidazole ring at CE1. The trimethylamino end of Diph699 packs over A1756 (*Figure 7*). The opposite surface of the tail is oriented toward the minor-groove side of the second base pair of the codon-anticodon helix (G6906:C6951). Thus, in comparison with the initiation state, the histidine-diphthamide tip of eEF2 replaces the codon-anticodon–like helix of PKI. The splitting of the interaction of A1755-A1756 and PKI is achieved by providing the histidine-diphthamine tip as a binding partner for both A1756 and the minor groove of the codon-anticodon helix (*Figure 7*).

Unlike in Structures II to V, the conformation of the eEF2 GTPase center in Structure I resembles that of a GTP-bound translocase (*Figure 5e*). In translational GTPases, switch loops I and II are involved in the GTPase activity (reviewed in *Voorhees and Ramakrishnan, (2013)*). Switch loop II (aa 105–110), which carries the catalytic H108 (H92 in *E. coli* EF-G; (*Cunha et al., 2013*; *Holtkamp et al., 2014*; *Koripella et al., 2015*; *Salsi et al., 2014*) is well resolved in all five structures. The histidine resides next to the backbone of G3028 of the sarcin-ricin loop and near the diphosphate of GDP (*Figure 5e*). By contrast, switch loop I (aa 50–70 in *S. cerevisiae* eEF2) is resolved only in Structure I (*Figure 5—figure supplement 2*). The N-terminal part of the loop (aa 50–60) is sandwiched between the tip of helix 14 ([415]CAAA[418]) of the 18S rRNA of the 40S subunit and helix A (aa 32–42) of eEF2 (*Figure 5d*). Bulged A416 interacts with the switch loop in the vicinity of D53. Next to GDP, the C-terminal part of the switch loop (aa 61–67) adopts a helical fold. As such, the conformations of SWI and the GTPase center in general are similar to those observed in ribosome-bound EF-Tu (*Voorhees et al., 2010*) and EF-G (*Pulk and Cate, 2013*; *Tourigny et al., 2013*; *Zhou et al., 2013*) in the presence of GTP analogs.

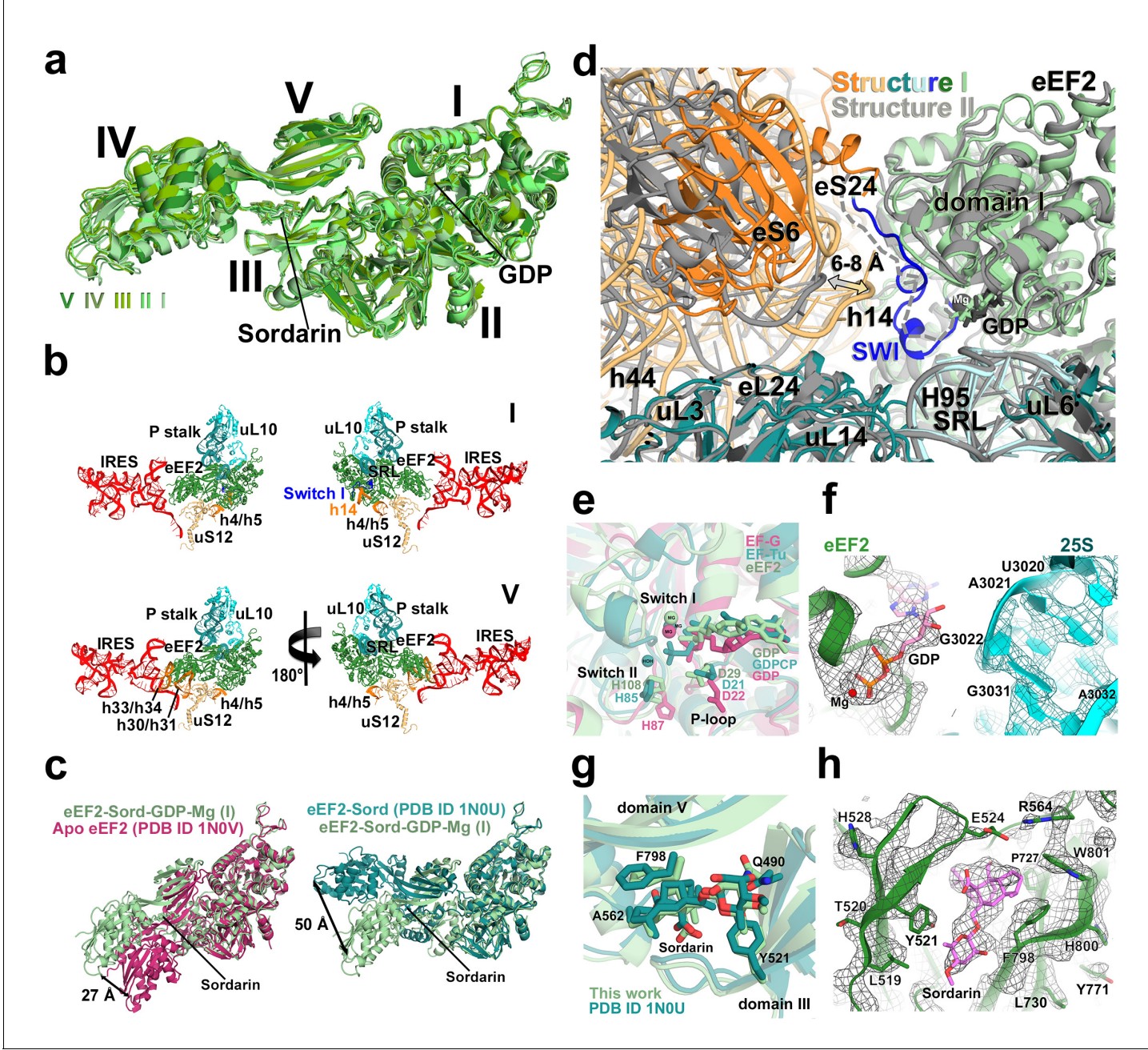

**Figure 5.** Conformations and interactions of eEF2. (**a**) Conformations of eEF2 in Structures I-V and domain organization of eEF2 are shown. Roman numerals denote eEF2 domains. Superposition was obtained by structural alignment of domains I and II. (**b**) Elements of the 80S ribosome in Structures I and V that contact eEF2. eEF2 is shown in green, IRES RNA in red, 40S subunit elements in orange, 60S in cyan/teal. (**c**) Comparison of conformations of eEF2•sordarin in Structure I (light green) with those of free apo-eEF2 (magenta) and eEF2•sordarin (teal) (*Jorgensen et al., 2003*). (**d**) Interactions of the GTPase domains with the 40S and 60S subunits in Structure I (colored in green/blue, eEF2; orange, 40S; cyan/teal, 60S) and in Structure II (gray). Switch loop I (SWI) in Structure I is in blue; dashed line shows the putative location of unresolved switch loop I in Structure II. Superposition was obtained by structural alignment of the 25S rRNAs. (**e**) Comparison of the GTP-like conformation of eEF2•GDP in Structure I (light green) with those of 70S-bound elongation factors EF-Tu•GDPCP (teal; *Voorhees et al. 2010*) and EF-G•GDP•fusidic acid (magenta; fusidic acid not shown; *Zhou et al., 2013*). (**f**) Cryo-EM density showing guanosine diphosphate bound in the GTPase center (green) next to the sarcin-ricin loop of 25S rRNA (cyan) of Structure II. (**g**) Comparison of the sordarin-binding sites in the ribosome-bound (light green; Structure II) and isolated eEF2 (teal; *Jorgensen et al., 2003*). (**h**) Cryo-EM density showing the sordarin-binding pocket of eEF2 (Structure II). Sordarin is shown in pink with oxygen atoms in red.

The following figure supplements are available for figure 5:

*Figure 5 continued on next page*

*Figure 5 continued*

**Figure supplement 1.** Elements of the 80S ribosome that contact eEF2 in Structures I through V.
**Figure supplement 2.** Cryo-EM density of the GTPase region in Structures I and II.

## Structure II reveals PKI between the body A and P sites and eEF2 partially advanced into the A site

In Structure II, relative to Structure I, PKI is further shifted along the 40S body, traversing ~4 Å toward the P site (*Figures 2e, f, and g*), while stacking on C1274 at the head A site. Thus, the intermediate position of PKI is possible due to a large swivel of the head relative to the body, which brings the head A site close to the body P site.

Domain IV of eEF2 is further entrenched in the A site by ~3 Å relative to the body and ~8 Å relative to the head, preserving its interactions with PKI. The decoding center residues A1755 and A1756 are rearranged to pack inside helix 44, making room for eEF2. This conformation of decoding center residues is also observed in the absence of A-site ligands (*Ogle et al., 2001*). The head interface of domain IV interacts with the 40S head (*Figure 6a*). Here, a positively charged surface of eEF2, formed by K613, R617 and R631 contacts the phosphate backbone of helix 33 (*Figures 6c*; see also *Figure 6—figure supplement 1*).

## Structure III represents a highly bent IRES with PKI captured between the head A and P sites

Consistent with the similar head swivels in Structure III and Structure II, relative positions of the 40S head A site and body P site remain as in Structure II. Among the five structures, the PKI domain is least ordered in Structure III and lacks density for SL3. The map allows placement of PKI at the body P site (*Figure 1—figure supplement 3*). Thus, in Structure III, PKI has translocated along the 40S body, but the head remains fully swiveled so that PKI is between the head A and P sites. Lower resolution of the map in this region suggests that PKI is somewhat destabilized in the vicinity of the body P site in the absence of stacking with the foundations of the head A site (C1274) or P site (U1191). The position of eEF2 is similar to that in Structure II.

## Structure IV represents a highly bent IRES with PKI partially accommodated in the P site

In Structure IV, the 40S subunit is almost non-rotated relative to the 60S subunit, and the 40S head is mid-swiveled. Unwinding of the head moves the head P-site residue U1191 and body P-site residue C1637 closer together, resulting in a partially restored 40S P site. Whereas C1637 forms a stacking platform for the last base pair of PKI, U1191 does not yet stack on PKI because the head remains partially swiveled. This renders PKI partially accommodated in the P site (*Figure 2g*).

Unwinding of the 40S head also positions the head A site closer to the body A site. This results in rearrangements of eEF2 interactions with the head, allowing eEF2 to advance further into the A site. To this end, the head-interacting interface of domain IV slides along the surface of the head by 5 Å. Helix A of domain IV is positioned next to the backbone of h34, with positively charged residues K613, R617 and R631 rearranged from the backbone of h33 (*Figure 6c*; see also *Figure 6—figure supplement 1*).

## Structure V represents an extended IRES with PKI fully accommodated in the P site and domain IV of eEF2 in the A site

In the nearly non-rotated and non-swiveled ribosome conformation in Structure V closely resembling that of the post-translocation 80S•2tRNA•mRNA complex (*Svidritskiy et al., 2014*), PKI is fully accommodated in the P site. The codon-anticodon–like helix is stacked on P-site residues U1191 and C1637 (*Figure 3d*), analogous to stacking of the tRNA-mRNA helix (*Figure 3e*).

A notable conformational change in eEF2 from that in the preceding Structures is visible in the position of domain III, which contacts uS12 (*Figure 6d*). In Structure V, protein uS12 is shifted along with the 40S body as a result of intersubunit rotation. In this position, uS12 forms extensive

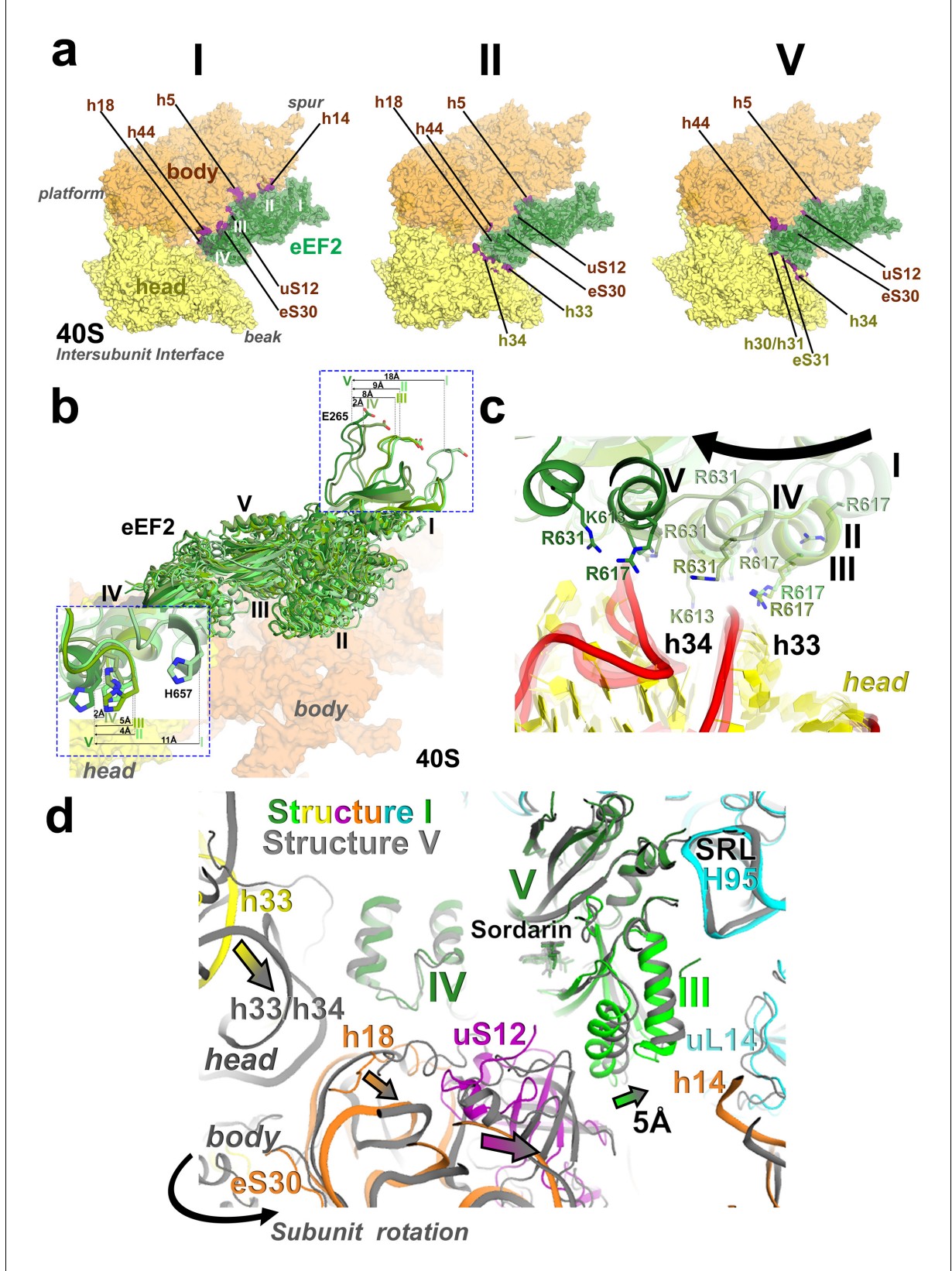

**Figure 6.** Interactions of eEF2 with the 40S subunit. (a) eEF2 (green) interacts only with the body in Structure I (eEF2 domains are labeled with roman numerals in white), and with both the head and body in Structures II through V. Colors are as in *Figure 1*, except for the 40S structural elements that

*Figure 6 continued on next page*

*Figure 6 continued*

contact eEF2, which are labeled and shown in purple. (**b**) Entry of eEF2 into the 40S A site, from Structure I through V. Distances to the A-site accommodated eEF2 (Structure V) are shown. The view was obtained by superpositions of the body domains of 18S rRNAs. (**c**) Rearrangements, from Structure I through V, of a positively charged cluster of eEF2 (K613, R617 and R631) positioned over the phosphate backbone of 18S helices 33 and 34, suggesting a role of electrostatic interactions in eEF2 diffusion over the 40S surface. (**d**) Shift of the tip of domain III of eEF2, interacting with uS12 upon reverse subunit rotation from Structure I to Structure V. Structure I colored as in *Figure 1*, except uS12, which is in purple; Structure V is in gray.

The following figure supplement is available for figure 6:

**Figure supplement 1.** Repositioning (sliding) of the positively-charged cluster of domain IV of eEF2 over the phosphate backbone (red) of the 18S helices 33 and 34.

interactions with eEF2 domains II and III. Specifically, the C-terminal tail of uS12 packs against the β-barrel of domain II, while the β-barrel of uS12 packs against helix A of domain III. This shifts the tip of helix A of domain III (at aa 500) by ~5 Å (relative to that in Structure I) toward domain I. Although domain III remains in contact with domain V, the shift occurs in the direction that could eventually disconnect the β-platforms of these domains.

Domain IV of eEF2 is fully accommodated in the A site. The first codon of the open reading frame is also positioned in the A site, with bases exposed toward eEF2 (*Figure 7*), resembling the conformations of the A-site codons in EF-G-bound 70S complexes. As in the preceding Structures, the histidine-diphthamide tip is bound in the minor groove of the P-site codon-anticodon helix. Diph699 slightly rearranges, relative to that in Structure I (*Figure 7*), and interacts with four out of six codon-anticodon nucleotides. The imidazole moiety stacks on G6907 (corresponding to nt 36 in the tRNA anticodon) and hydrogen bonds with O2' of G6906 (nt 35 of tRNA). The amide at the diphthamide end interacts with N2 of G6906 and O2 and O2' of C6951 (corresponding to nt 2 of the codon). The trimethylamino-group is positioned over the ribose of C6952 (codon nt 3).

## Discussion

### IRES translocation mechanism

In this work we have captured the structures of the TSV IRES, whose PKI samples positions between the A and P sites (Structures I–IV), as well as in the P site (Structure V). We propose that together with the previously reported initiation state (*Koh et al., 2014*), these structures represent the trajectory of eEF2-induced IRES translocation (shown as an animation in http://labs.umassmed.edu/korostelevlab/msc/iresmovie.gif and *Video 1*). Our structures reveal previously unseen intermediate states of eEF2 or EF-G engagement with the A site, providing the structural basis for the mechanism of translocase action. Furthermore, they provide insight into the mechanism of eEF2•GTP association with the pre-translocation ribosome and eEF2•GDP dissociation from the post-translocation ribosome, also delineating the mechanism of translation inhibition by the antifungal drug sordarin. In summary, the reported ensemble of structures substantially enhances our understanding of the translocation mechanism, including that of tRNAs as discussed below.

Translocation of the TSV IRES on the 40S subunit globally resembles a step of an inchworm (*Figure 4*; see also *Figure 3—figure supplement 2*). At the start (initiation state), the IRES adopts an extended conformation (extended inchworm). The front 'legs' (SL4 and SL5) of the 5'-domain (front end) are attached to the 40S head proteins uS7, uS11 and eS25 (*Figure 3—figure supplement 2*). PKI, representing the hind end, is bound in the A site. In the first sub-step (Structures I to IV), the hind end advances from the A to the P site and approaches the front end, which remains attached to the 40S surface. This shortens the distance between PKI and SL4 by up to 20 Å relative to the initiating IRES structure, resulting in a bent IRES conformation (bent inchworm). Finally (Structures IV to V), as the hind end is accommodated in the P site, the front 'legs' advance by departing from their initial binding sites. This converts the IRES into an extended conformation, rendering the inchworm prepared for the next translocation step. Notably, at all steps, the head of the IRES inchworm (L1.1 region) is supported by the mobile L1 stalk. In the post-translocation CrPV IRES structure (*Muhs et al., 2015*), the 5'-domain similarly protrudes between the subunits and interacts with the

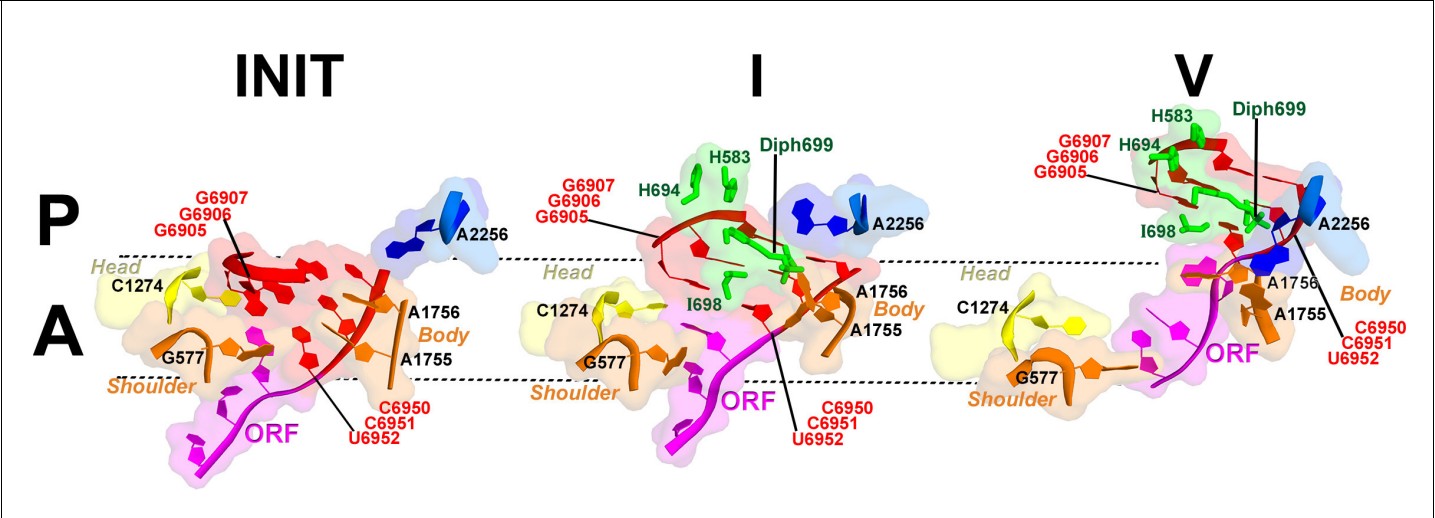

**Figure 7.** Interactions of the residues at the eEF2 tip with the decoding center of the IRES-bound ribosome. Key elements of the decoding center of the 'locked' initiation structure (*Koh et al., 2014*), 'unlocked' Structure I, and post-translocation Structure V (this work) are shown. The histidine-diphthamide tip of eEF2 is shown in green. The codon-anticodon-like helix of PKI is shown in red, the downstream first codon of the ORF in magenta. Nucleotides of the 18S rRNA body are in orange and head in yellow; 25S rRNA nucleotide A2256 is blue. **A** and **P** sites are schematically demarcated by dotted lines.

L1 stalk, as in the initiation state for this IRES (*Fernandez et al., 2014*). This underlines structural similarity for the TSV and CrPV IRES translocation mechanisms.

Upon translocation, the GCU start codon is positioned in the A site (Structure V), ready for interaction with Ala-tRNA$^{Ala}$ upon eEF2 departure. Recent studies have shown that in some cases a fraction of IGR IRES-driven translation results from an alternative reading frame, which is shifted by one nucleotide relative to the normal ORF (*Au and Jan, 2012*; *Ren et al., 2012*; *2014*; *Wang and Jan, 2014*). One of the mechanistic scenarios (discussed in *Ren et al., 2014*) involves binding of the first aminoacyl-tRNA to the post-translocated IRES mRNA frame shifted by one nucleotide (predominantly a +1 frame shift). In our structures, the IRES presents to the decoding center a pre-translocated or fully translocated ORF, rather than a +1 (more translocated) ORF, suggesting that eEF2 does not induce a highly populated fraction of +1 shifted IRES mRNAs. It is likely that alternative frame setting occurs following eEF2 release and that this depends on transient displacement of the start codon in the decoding center, allowing binding of the corresponding amino acyl-tRNA to an off-frame codon. Further structural studies involving 80S•IRES•tRNA complexes are necessary to understand the mechanisms underlying alternative reading frame selection.

The presence of several translocation complexes in a single sample suggests that the structures represent equilibrium states of forward and reverse translocation of the IRES, which interconvert among each other. This is consistent with the observations that the intergenic IRESs are prone to reverse translocation. Specifically, biochemical toe-printing studies in the presence of eEF2•GTP identified IRES in a non-translocated position unless eEF1a•aa-tRNA is also present (*Jan et al., 2003*; *Pestova and Hellen, 2003*; *Yamamoto et al., 2007*). These findings indicate that IRES translocation by eEF2 is futile: the IRES returns to the A site upon releasing eEF2•GDP unless an aminoacyl tRNA enters the A site and blocks IRES back-translocation. This contrasts with the post-translocated 2tRNA•mRNA complex, in which the classical P and E-site tRNAs are stabilized in the non-rotated ribosome after translocase release (*Chen et al., 2013a*; *Ermolenko and Noller, 2011*). Thus, the meta-stability of the post-translocation IRES is likely due to the absence of stabilizing structural features present in the 2tRNA•mRNA complex. In the initiation state, the IRES resembles a pre-translocation 2tRNA•mRNA complex (*Brilot et al., 2013*) reduced to the A/P-tRNA anticodon-stem loop and elbow in the A site and the P/E-tRNA elbow contacting the L1 stalk. Because the anticodon-stem loop of the A-tRNA is sufficient for translocation completion (*Joseph and Noller, 1998*; *Studer et al., 2003*), we ascribe the meta-stability of the post-translocation IRES to the absence of

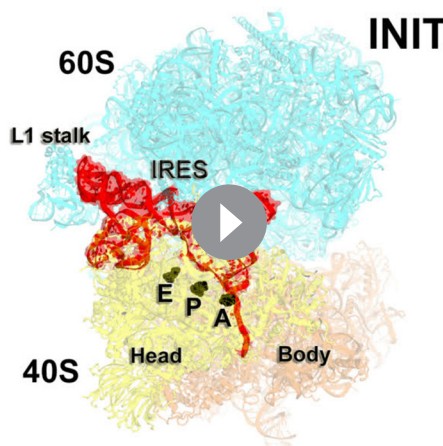

**Video 1.** Animation showing the transition from the initiation 80S•TSV IRES structures (Koh et al., 2014) to eEF2-bound Structures I through V (this work). Four views (scenes) are shown: (1) A view down the intersubunit space, with the head of the 40S subunit oriented toward a viewer, as in *Figure 1a*; (2) A view at the solvent side of the 40S subunit, with the 40S head shown at the top, as in *Figure 2—figure supplement 1*; (3) A view down at the subunit interface of the 40S subunit; (4) A close-up view of the decoding center (A site) and the P site, as in *Figure 2g*. Each scene is shown twice. Colors are as in *Figure 1*. In scenes 1, 2 and 3, nucleotides C1274, U1191 of the 40S head and G904 of the 40S platform are shown in black to denote the A, P and E sites, respectively. In scene 4, C1274 and U1191 are labeled and shown in yellow; G577, A1755 and A1756 of the 40S body A site and C1637 of the body P site are labeled and shown in orange.

the P/E-tRNA elements, either the ASL or the acceptor arm, or both. Furthermore, interactions of SL4 and SL5 with the 40S subunit likely contribute to stabilization of pre-translocation structures.

## Partitioned roles of 40S subunit rearrangements

Our structures delineate the mechanistic functions for intersubunit rotation and head swivel in translocation. These functions are partitioned. Specifically, intersubunit rotation allows eEF2 entry into the A site, while the head swivel mediates PKI translocation. Various degrees of intersubunit rotation have been observed in cryo-EM studies of the 80S•IRES initiation complexes (*Fernandez et al., 2014*; *Koh et al., 2014*). This suggests that the subunits are capable of spontaneous rotation, as is the case for tRNA-bound pre-translocation complexes (*Cornish et al., 2008*). The pre-translocation Structure I with eEF2 least advanced into the A site adopts a fully rotated conformation. Reverse intersubunit rotation from Structure I to V shifts the translocation tunnel (the tunnel between the A, P and E sites) toward eEF2, which is rigidly attached to the 60S subunit. This allows eEF2 to move into the A site. As such, reverse intersubunit rotation facilitates full docking of eEF2 in the A site.

Because the histidine-diphthamide tip of eEF2 (H583, H694 and Diph699) attaches to the codon-anticodon-like helix of PKI, eEF2 appears to directly force PKI out of the A site. The head swivel allows gradual translocation of PKI to the P site, first with respect to the body and then to the head. The fully swiveled conformations of Structures II and III represent the mid-point of translocation, in which PKI relocates between the head A site and body P site. We note that such mid-states have not been observed for 2tRNA•mRNA, but their formation can explain the formation of subsequent pe/E hybrid (*Ratje et al., 2010*) and ap/P chimeric structures (*Ramrath et al., 2013*; *Zhou et al., 2014*) (*Figure 1—figure supplement 1*). Reverse swivel from Structure III to V brings the head to the non-swiveled position, restoring the A and P sites on the small subunit.

## The functions of eEF2 in translocation

To our knowledge, our work provides the first high-resolution view of the dynamics of a ribosomal translocase that is inferred from an ensemble of structures sampled under uniform conditions. The structures, therefore, offer a unique opportunity to address the role of the elongation factors during translocation. Translocases are efficient enzymes. While the ribosome itself has the capacity to translocate in the absence of the translocase, spontaneous translocation is slow (*Cukras et al., 2003*; *Ermolenko et al., 2013*; *Gavrilova et al., 1976*; *Gavrilova and Spirin, 1972*; *Pestka, 1968*). EF-G enhances the translocation rate by several orders of magnitude, aided by an additional 2- to 50-fold boost from GTP hydrolysis (*Ermolenko and Noller, 2011*; *Rodnina et al., 1997*). Due to the lack of structures of translocation intermediates, the mechanistic role of eEF2/EF-G is not fully understood.

The 80S•IRES•eEF2 structures reported here suggest two main roles for eEF2 in translocation. As discussed above, the first role is to directly shift PKI out of the A site upon spontaneous reverse intersubunit rotation. In our structures, the tip of domain IV docks next to PKI, with diphthamide 699

fit into the minor groove of the codon-anticodon-like helix of PKI (*Figure 7*). This arrangement rationalizes inactivation of eEF2 by diphtheria toxin, which catalyzes ADP-ribosylation of the diphthamide (reviewed in *Collier, 2001*). The enzyme ADP-ribosylates the NE2 atom of the imidazole ring, which in our structures interacts with the first two residues of the anticodon-like strand of PKI. The bulky ADP-ribosyl moiety at this position would disrupt the interaction, rendering eEF2 unable to bind to the A site (*Nygard and Nilsson, 1990*) and/or stalled on ribosomes in a non-productive conformation (*Bermek, 1976*; *Davydova and Ovchinnikov, 1990*).

As eEF2 shifts PKI toward the P site in the course of reverse intersubunit rotation, the 60S-attached translocase migrates along the surface of the 40S subunit, guided by electrostatic interactions. Positively-charged patches of domains II and III (R391, K394, R433, R510) and IV (K613, R617, R609, R631, K651) slide over rRNA of the 40S body (h5) and head (h18 and h33/h34), respectively. The Structures reveal hopping of the positive clusters over rRNA helices. For example, between Structures II and V, the K613/R617/R631 cluster of domain IV hops by ∼19 Å (for Cα of R617) from the phosphate backbone of h33 (at nt 1261–1264) to that of the neighboring h34 (at nt 1442–1445). Thus, sliding of eEF2 involves reorganization of electrostatic, perhaps isoenergetic interactions, echoing those implied in extraordinarily fast ribosome inactivation rates by the small-protein ribotoxins (*Korennykh et al., 2006*) and in fast protein association and diffusion along DNA (*Givaty and Levy, 2009*; *Gorman et al., 2010*; *Halford, 2009*).

Comparison of our structures with the 80S•IRES initiation structure reveals the structural basis for the second key function of the translocase: 'unlocking' of intrasubunit rearrangements that are required for step-wise translocation of PKI on the small subunit. The unlocking model of the ribosome•2tRNA•mRNA pre-translocation complex has been proposed decades ago (*Spirin, 1969*) and functional requirement of the translocase in this process has been implicated (*Savelsbergh et al., 2003*). However, the structural and mechanistic definitions of the locked and unlocked states have remained unclear, ranging from the globally distinct ribosome conformations (*Valle et al., 2003*) to unknown local rearrangements, *e.g.* those in the decoding center (*Taylor et al., 2007*). FRET data indicate that translocation of 2tRNA•mRNA on the 70S ribosome requires a forward-and-reverse head swivel (*Guo and Noller, 2012*), which may be related to the unlocking phenomenon. Whereas intersubunit rotation of the pre-translocation complex occurs spontaneously, the head swivel is induced by the eEF2/EF-G translocase, consistent with requirement of eEF2 for unlocking. Structural studies revealed large head swivels in various 70S•tRNA•EF-G (*Ramrath et al., 2013*; *Ratje et al., 2010*; *Zhou et al., 2013*) and 80S•tRNA•eEF2 (*Taylor et al., 2007*) complexes, but not in 'locked' complexes with the A site occupied by the tRNA in the absence of the translocase (*Agirrezabala et al., 2008*; *Demeshkina et al., 2012*; *Jenner et al., 2010*; *Selmer et al., 2006*).

Our structures suggest that eEF2 induces head swivel by 'unlocking' the head-body interactions (*Figure 7*). Binding of the ASL to the A site is known from structural studies of bacterial ribosomes to result in 'domain closure' of the small subunit, *i.e.* closer association of the head, shoulder and body domains (*Ogle et al., 2001*). The domain closure 'locks' cognate tRNA in the A site *via* stacking on the head A site (C1274 in *S. cerevisiae* or C1054 in *E. coli*) and interactions with the body A-site nucleotides A1755 and A1756 (A1492 and A1493 in *E. coli*). This 'locked' state is identical to that observed for PKI in the 80S•IRES initiation structures in the absence of eEF2 (*Fernandez et al., 2014*; *Koh et al., 2014*). Structure I demonstrates that at an early pre-translocation step, the histidine-diphthamide tip of eEF2 is wedged between A1755 and A1756 and PKI. This destabilization allows PKI to detach from the body A site upon spontaneous reverse 40S body rotation, while maintaining interactions with the head A site. Destabilization of the head-bound PKI at the body A site thus allows mobility of the head relative to the body. The histidine-diphthamide-induced disengagement of PKI from A1755 and A1756 therefore provides the structural definition for the 'unlocking' mode of eEF2 action.

In summary, our structures are consistent with a model of eEF2-induced translocation in which both PKI and eEF2 passively migrate into the P and A site, respectively, during spontaneous 40S body rotation and head swivel, the latter being allowed by 'unlocking' of the A site by eEF2. Observation of different PKI conformations sampling a range of positions between the A and P sites in the presence of eEF2•GDP implies that thermal fluctuations of the 40S head domain are sufficient for translocation along the energetically flat trajectory.

## Insights into eEF2 association with and dissociation from the ribosome

The conformational rearrangements in eEF2 from Structure I through Structure V provide insights into the mechanisms of eEF2 association with the pre-translocation ribosome and dissociation from the post-translocation ribosome. In all five structures, the GTPase domain is attached to the P stalk and the sarcin-ricin loop. In the fully-rotated pre-translocation-like Structure I, an additional interaction exists. Here, switch loop I interacts with helix 14 ($^{415}$CAAA$^{418}$) of the 18S rRNA. This stabilization renders the GTPase center to adopt a GTP-bound conformation, similar to those observed in other translational GTPases in the presence of GTP analogs (*Pulk and Cate, 2013*; *Tourigny et al., 2013*; *Voorhees et al., 2010*; *Zhou et al., 2013*) and in the 80S•eEF2 complex bound with a transition-state mimic GDP•AlF$^{4-}$ (*Sengupta et al., 2008*). The switch loop contacts the base of A416 (invariable A344 in *E. coli* and A463 in *H. sapiens*). Mutations of residues flanking A344 in *E. coli* 16S rRNA modestly inhibit translation but do not specifically affect EF-G-mediated translocation (*Sahu et al., 2012*). However, the effect of A344 mutation on translation was not addressed in that study, leaving the question open whether this residue is critical for eEF2/EF-G function. The interaction between h14 and switch loop I is not resolved in Structures II to V, in all of which the small subunit is partially rotated or non-rotated, so that helix 14 is placed at least 6 Å farther from eEF2 (*Figure 5d*). We conclude that unlike other conformations of the ribosome, the fully rotated 40S subunit of the pre-translocation ribosome provides an interaction surface, complementing the P stalk and SRL, for binding of the GTP-bound translocase. This structural basis rationalizes the observation of transient stabilization of the rotated 70S ribosome upon EF-G•GTP binding and prior to translocation (*Chen et al., 2013a*; *Ermolenko and Noller, 2011*; *Fei et al., 2008*; *Pan et al., 2007*; *Spiegel et al., 2007*).

The least rotated conformation of the post-translocation Structure V suggests conformational changes that may trigger eEF2 release from the ribosome at the end of translocation. The most pronounced inter-domain rearrangement in eEF2 involves movement of domain III. In the rotated or mid-rotated Structures I through III, this domain remains rigidly associated with domain V and the N-terminal superdomain and does not undergo noticeable rearrangements. In Structure V, however, the tip of helix A of domain III is displaced toward domain I by ~5 Å relative to that in mid-rotated or fully rotated structures. This displacement is caused by the 8 Å movement of the 40S body protein uS12 upon reverse intersubunit rotation from Structure I to V (*Figure 6d*). We propose that the shift of domain III by uS12 initiates intra-domain rearrangements in eEF2, which unstack the β-platform of domain III from that of domain V. This would result in a conformation characteristic of free eEF2 and EF-G in which the β-platforms are nearly perpendicular (*Czworkowski et al., 1994*; *Evarsson et al., 1994*; *Jorgensen et al., 2003*). As we discuss below, Structure V captures a 'pre-unstacking' state due to stabilization of the interface between domains III and V by sordarin.

## Sordarin stabilizes GDP-bound eEF2 on the ribosome

Sordarin is a potent antifungal antibiotic that inhibits translation. Based on biochemical experiments, two alternative mechanisms of action were proposed: sordarin either prevents eEF2 departure by inhibiting GTP hydrolysis (*Dominguez et al., 1999*) or acts after GTP hydrolysis (*Justice et al., 1998*). Although our complex was assembled using eEF2•GTP, density maps clearly show GDP and Mg$^{2+}$ in each structure (*Figure 5g*). Our structures therefore indicate that sordarin stalls eEF2 on the ribosome in the GDP-bound form, *i.e.* following GTP hydrolysis and phosphate release.

The mechanism of stalling is suggested by comparison of pre-translocation and post-translocation structures in our ensemble. In all five structures, sordarin is bound between domains III and V of eEF2, stabilized by hydrophobic interactions identical to those in the isolated eEF2•sordarin complex (*Figures 5g and h*). In the nearly non-rotated post-translocation Structure V, the tip of domain III is shifted, however the interface between domains III and V remains unchanged, suggesting strong stabilization of this interface by sordarin. We note that Structure V is slightly more rotated than the 80S•2tRNA•mRNA complex in the absence of eEF2•sordarin, implying that sordarin interferes with the final stages of reverse rotation of the post-translocation ribosome. We propose that sordarin acts to prevent full reverse rotation and release of eEF2•GDP by stabilizing the interdomain interface and thus blocking uS12-induced disengagement of domain III from domain V.

## Implications for tRNA and mRNA translocation during translation

Because translocation of tRNA must involve large-scale dynamics, this step has long been regarded as the most puzzling step of translation. Intersubunit rearrangements and tRNA hybrid states have been proposed to play key roles half a century ago (*Bretscher, 1968*; *Spirin, 1969*). Despite an impressive body of biochemical, fluorescence and structural data accumulated since then, translocation remains the least understood step of elongation (*Joseph, 2003*; *Ling and Ermolenko, 2016*; *Voorhees and Ramakrishnan, 2013*). The structural understanding of ribosome and tRNA dynamics has been greatly aided by a wealth of X-ray and cryo-EM structures (reviewed in *Agirrezabala and Valle, 2015*; *Dunkle and Cate, 2010*; *Korostelev et al., 2008*). However, visualization of the eEF2/EF-G-induced translocation is confined to very early pre-EF-G-entry states (*Brilot et al., 2013*; *Lin et al., 2015*) and late (almost translocated or fully translocated) states (*Gao et al., 2009*; *Ramrath et al., 2013*; *Zhou et al., 2014*), leaving most of the path from the A to the P site uncharacterized (*Figure 1—figure supplement 1*).

Our study provides new insights into the structural understanding of tRNA translocation. First, we propose that tRNA and IRES translocations occur *via* the same general trajectory. This is evident from the fact that ribosome rearrangements in translocation are inherent to the ribosome (*Agirrezabala et al., 2008*; *Cornish et al., 2008*; *Gavrilova et al., 1976*; *Julián et al., 2008*) and likely occur in similar ways in both cases. Furthermore, the step-wise coupling of ribosome dynamics with IRES translocation is overall consistent with that observed for 2tRNA•mRNA translocation in solution. For example, fluorescence and biochemical studies revealed that the early pre-translocation EF-G-bound ribosomes are fully rotated (*Chen et al., 2013a*; *Ermolenko and Noller, 2011*; *Spiegel et al., 2007*) and translocation of the tRNA-mRNA complex occurs during reverse rotation of the small subunit (*Ermolenko and Noller, 2011*), coupled with head swivel (*Guo and Noller, 2012*). The sequence of ribosome rearrangements during IRES translocation also agrees with that inferred from 70S•EF-G structures, including those in which the A-to-P-site translocating tRNA was not present. Specifically, an earlier translocation intermediate ribosome (TIpre) was proposed to adopt a rotated (7–9°) body and a partly rotated head (5–7.5°) (*Chen et al., 2013b*; *Ratje et al., 2010*; *Tourigny et al., 2013*), in agreement with the conformation of our Structure I. The most swiveled head (18–21°) was observed in a mid-rotated ribosome (3–5°) of a later translocation intermediate TIpost (*Ramrath et al., 2013*; *Ratje et al., 2010*), similar to the conformation of our Structure III. Overall, these correlations suggest that the intermediate locations of the elusive A-to-P-site translocating tRNA are similar to those of PKI in our structures.

Second, the structures clarify the structural basis of the often-used but structurally undefined terms 'locking' and 'unlocking' with respect to the pre-translocation complex (*Figure 6f*). We deem the pre-translocation complex locked, because the A-site bound ASL-mRNA is stabilized by interactions with the decoding center (*Ogle et al., 2001*). These interactions are maintained for the classical- and hybrid-state tRNAs in the spontaneously sampled non-rotated and rotated ribosomes, respectively (*Ermolenko et al., 2007*; *Spiegel et al., 2007*). Unlocking involves separation of the codon-anticodon helix from the decoding center residues by the protruding tip of eEF2/EF-G (*Figure 7*), occurring in the fully rotated ribosome at an early pre-translocation step. This unlatches the head, allowing creation of hitherto elusive intermediate tRNA positions during spontaneous reverse body rotation.

Third, our findings uncover a new role of the head swivel. Previous studies showed that this movement widens the constriction ('gate') between the P and E sites, thus allowing the P-tRNA passage to the E site (*Schuwirth et al., 2005*; *Spahn et al., 2004a*; *Taylor et al., 2007*; *Zhou et al., 2014*). In addition to the 'gate-opening' role, we now show that the head swivel brings the head A site to the body P site, allowing a step-wise conveying of the codon-anticodon helix between the A and P sites.

Finally, the similar populations of particles (within a 2X range) in our 80S•IRES•eEF2 reconstructions (*Figure 1—figure supplement 2*) suggest that the intermediate translocation states sample several energetically similar and interconverting conformations. This is consistent with the idea of a rather flat energy landscape of translocation, suggested by recent work that measured mechanical work produced by the ribosome during translocation (*Liu et al., 2014*). Our findings implicate, however, that the energy landscape is not completely flat and contains local minima for transient positions of the codon-anticodon helix between the A and P sites. The shift of the PKI with respect to the body occurs during forward head swivel in two major sub-steps of ~4 Å each (initiation complex

to I, and I to II), after which PKI undergoes small shifts to settle in the body P site in Structures III, IV and V (*Figure 2—source data 1*). Movement of PKI relative to the head occurs during the subsequent reverse swivel in three 3–7 Å sub-steps (II to III to IV to V). It is possible that additional meta-stable but less populated states exist between the conformations we observe. We note that four of our near-atomic resolution maps comprised ~30,000 particles each, the minimum number required for a near-atomic-resolution reconstruction of the ribosome (*Bai et al., 2013*). A larger data set will therefore be necessary to reveal additional sub-states.

## Concluding remarks

### Translation of viral mRNA

Our work sheds light on the dynamic mechanism of cap-independent translation by IGR IRESs, tightly coupled with the universally conserved dynamic properties of the ribosome. The cryo-EM structures demonstrate that the TSV IRES structurally and dynamically represents a chimera of the 2tRNA•mRNA translocating complex (A/P-tRNA • P/E-tRNA • mRNA). Like in the 2tRNA•mRNA translocating complex in which the two tRNAs move independently of each other, the PKI domain moves relative to the 5′-domain, causing the IRES to undergo an inchworm-walk translocation. A large structural difference between the IRES and the 2tRNA•mRNA complex exists, however, in that the IRES lacks three out of six tRNA-like domains involved in tRNA translocation. This difference likely accounts for the inefficient translocation of the IRES, which is difficult to stabilize in the post-translocation state and therefore is prone to reverse translocation. Although structurally handicapped, the TSV IRES manages to translocate by employing ribosome dynamics that are remarkably similar to that in 2tRNA•mRNA translocation. The uniformity of ribosome dynamics underscores the idea that translocation is an inherent and structurally-optimized property of the ribosome, supported also by translocation activity in the absence of the elongation factor. This property is rendered by the relative mobility of the three major building blocks, the 60S subunit and the 40S head and body, assisted by ligand-interacting extensions including the L1 stalk and the P stalk. Intergenic IRESs, in turn, represent a striking example of convergent molecular evolution. Viral mRNAs have evolved to adopt an atypical structure to employ the inherent ribosome dynamics, to be able to hijack the host translational machinery in a simple fashion.

### Ensemble cryo-EM

Our current understanding of macromolecular machines, such as the ribosome, is often limited by a gap between biophysical/biochemical studies and structural studies. For example, Förster resonance energy transfer can provide insight into the macromolecular dynamics of an assembly at the single-molecule level but is limited to specifically labeled locations within the assembly. High-resolution crystal structures, on the other hand, can provide static images of an assembly, and the structural dynamics can only be inferred by comparing structures that are usually obtained in different experiments and under different, often non-native, conditions. Cryo-EM offers the possibility of obtaining integrated information of both structure and dynamics as demonstrated in lower-resolution studies of bacterial ribosome complexes (*Agirrezabala et al., 2008*; *Fischer et al., 2010*; *Julián et al., 2008*). Recent technical advances, including direct electron detectors and image processing software (*Cheng et al., 2015*), have significantly improved the resolution at which such studies can be performed. The increased resolution, need for larger datasets and more sophisticated algorithms have also led to a massive increase in the computational power required to process the data. The available computing infrastructure and computational efficiency have therefore become deciding factors in how many different structural states can be resolved. This is presumably one of the reasons why most recent studies of ribosome complexes have focused on a single high-resolution structure despite the non-uniform local resolution of the maps that likely reflects structural heterogeneity. The computational efficiency of FREALIGN (*Lyumkis et al., 2013*) has allowed us to classify a relatively large dataset (1.1 million particles) into 15 classes (*Figure 1—figure supplement 2*) and obtain eight near-atomic-resolution structures from it. The classification, which followed an initial alignment of all particles to a single reference, required about 130,000 CPU hours or about five to six full days on a 1000-CPU cluster. While it would clearly be impractical to perform this type of analysis on a desktop computer, one may extrapolate using Moore's law (*Moore, 1965*) that such practice will become routine in less than ten years. Therefore, cryo-EM has the potential to become a standard tool for

uncovering detailed dynamic pathways of complex macromolecular machines. A particularly exciting application will be to infer the high-resolution temporal trajectory of a pathway from an ensemble of equilibrium states in a single sample, as described in our work, together with an analysis of samples quenched at different time points of the reaction (*Chen et al., 2015*; *Fischer et al., 2010*; *Shaikh et al., 2014*).

## Materials and methods

### *S. cerevisiae* 80S ribosome preparation

80S ribosomes used in this study were prepared from *Saccharomyces cerevisiae* strain W303 as described previously (*Ben-Shem et al., 2011*; *Koh et al., 2014*). To obtain ribosomal subunits, purified 80S was incubated in dissociation buffer (20 mM HEPES·KOH (pH 7.5), 0.5 M KCl, 2.5 mM magnesium acetate, 2 mM dithiothreitol (DTT), and 0.5 U/μl RNasin) for 1 hr at 4°C. The dissociated subunits were then layered on sucrose gradients (10% to 30% sucrose) in the dissociation buffer and centrifuged for 15 hr at 22,000 rpm in an SW32 rotor. Fractions corresponding to 40S and 60S subunits were pooled and buffer-exchanged to subunit storage buffer containing 50 mM Tris (pH7.5), 20 mM $MgCl_2$, 100 mM KCl, and 2 mM DTT. Purified subunits were flash-frozen in liquid nitrogen and stored in aliquots at –80°C.

### Taura syndrome virus IRES preparation

Plasmid pUC57 (Genscript) containing the synthetic DNA encoding for nucleotides 6741–6990 of the TSV mRNA sequence was used to amplify the 250-nucleotide fragment by PCR. This DNA fragment (TSV IRES RNA) served as a template for *in vitro* transcription. The transcription reaction was incubated for 4 hr at 37°C, and the resulting transcription product was treated with DNase I for 30 mins at 37°C. The RNA was then extracted with acidic phenol/chloroform, gel-purified, and then ethanol precipitated with 100% ethanol, followed by an 80% ethanol wash. The resulting RNA pellet was air-dried at room temperature and suspended in RNase-free water. The TSV IRES transcription product was folded in 1X IRES refolding buffer (20 mM Potassium acetate pH 7.5 and 5 mM $MgCl_2$), incubated at 65°C for 10 min and cooled down gradually at room temperature, prior to complex preparation for cryo-EM study.

### *S. cerevisiae* eEF2 purification

C-terminally $His_6$-tagged eEF2 was produced in yeast TKY675 cells obtained from Terri Goss Kinzy. Yeast cells were grown in 4 liters of YPD media at 27°C and 160 rpm, to $A_{600}$=1.5. Yeast cell pellet (∼5 g) was obtained by centrifugation and re-suspended in 20 ml of the lysis buffer (50 mM potassium phosphate pH 7.6, 1 M KCl, 1% Tween 20, 10% Glycerol, 10 mM imidazole, 0.2 mM PMSF, 1 mM DTT, and 1 tablet of Roche miniComplete protease inhibitor). The suspension was lysed with microfluidizer at 25,000 psi at 4°C, and then clarified by centrifugation twice at 30,000 × *g* for 20 min. The supernatant was subjected to Ni-NTA affinity chromatography using the AKTAexplorer 100 system (GE Healthcare). After lysate application onto the column, the column was washed with a five-column volume of wash buffer (50 mM potassium phosphate pH 7.6, 1 M KCl, 1% Tween 20, 10% Glycerol, 20 mM imidazole, 0.2 mM PMSF and 1 mM DTT). A gradient elution method was used to reach the final imidazole concentration of 250 mM. Eluted fractions were buffer-exchanged into buffer A (30 mM HEPES·KOH (pH 7.5), 5% glycerol, 65 mM ammonium chloride, 7 mM β–mercaptoethanol and 1 tablet of miniComplete protease inhibitor) for HiTrap SP Sepharose High Performance cation-exchange chromatography (GE Healthcare). A gradient elution method was used to reach the final salt concentration of 1 M KCl in buffer A over the 20-column volume (100 ml). The peak fraction was concentrated and buffer-exchanged into buffer A, which is also the buffer used for the subsequent size-exclusion chromatography employing Superdex 200 (GE Healthcare). Fractions corresponding to the eEF2 peak were concentrated and stored in aliquots at -20°C.

### 80S•TSV IRES•eEF2•GTP•sordarin complex preparation

The IRES-eEF2-ribosome complex was assembled in two steps. First, refolded TSV IRES RNA (8 μM - all concentrations are specified for the final complex) was incubated with the yeast 40S small subunit (0.8 μM) for 15 min at 30°C, in the buffer containing 45 mM HEPES·KOH (pH 7.5), 10 mM $MgCl_2$,

100 mM KCl, 2.5 mM spermine and 2 mM β–mercaptoethanol. The 60S subunit (0.8 μM) was then added and incubated for 15 min at 30°C. Subsequently, eEF2 (5 μM), sordarin (800 μM) and GTP (1 mM) were added and incubated for 15 min at 30°C. The solution was then incubated on ice for 10 min and flash-frozen in liquid nitrogen.

## Cryo-EM specimen preparation

Quantifoil Cu 200 mesh grids (SPI Supplies, West Chester, PA) were coated with a thin layer of carbon and glow discharged for 45 s at 25 mA. 3 μL of sample with a concentration of ∼0.1 μM was applied to the grid, incubated for 30 s and plunged into liquid ethane using an FEI Vitrobot Mark 2 (FEI Company, Hillsboro, OR) after blotting for 3 s at 4°C and ∼85% relative humidity.

## Electron microscopy

Cryo-EM data were collected in movie mode on an FEI Krios microscope (FEI Company, Hillsboro, OR) operating at 300 kV and equipped with a K2 Summit direct detector (Gatan Inc., Pleasanton, CA) operating in super-resolution mode with pixel size of 0.82 Å per super-resolution pixel. Each movie consisted of 50 frames collected over 18.8 s with an exposure per frame of 1.4 e-/Å2 as shown by Digital Micrograph (Gatan Inc., Pleasanton, CA), giving a total exposure of 70 e-/Å2. The defocus ranged between ∼0.7 to ∼2.5 μm underfocus.

## Image processing

The gain-corrected super-resolution movie frames were corrected for anisotropic magnification using bilinear interpolation (*Grant and Grigorieff, 2015a*). The frames were downsampled by Fourier cropping to a pixel size of 1.64 Å. The downsampled frames were then motion-corrected and exposure filtered using Unblur (*Grant and Grigorieff, 2015b*). The image defocus was determined using CTFFIND4 (*Rohou and Grigorieff, 2015*) on non-exposure-filtered images and images with excessive motion, low contrast, ice contamination or poor power spectra were removed based on visual inspection using TIGRIS (http://tigris.sourceforge.net/). 50 particles were picked manually using TIGRIS, summed and rotationally averaged to serve as a reference for correlation-based particle picking in IMAGIC (*van Heel et al., 1996*). 1,105,737 two-dimensional images of ribosomes (termed 'particles') were picked automatically, extracted into 256 x 256 boxes and converted to MRC/CCP4 format with a corresponding list of micrograph numbers and defocus values for input to FREALIGN v9 (*Lyumkis et al., 2013*).

The summary of procedures resulting in 3D cryo-EM maps is presented on *Figure 1—figure supplement 2*. FREALIGN v9 was used for refinement, classification and 3D reconstruction of all ribosome structures. Initial particle alignments were obtained by performing an angular grid search (FREALIGN mode 3) with a density map calculated from the atomic model of the non-rotated 80S ribosome bound with 2 tRNAs (PDB: 3J78 *Svidritskiy et al., 2014*). For this search, the resolution was limited to 20 Å and the resolution of the resulting reconstruction was 3.6 Å, as determined by the FSC = 0.143 threshold criterion (*Rosenthal and Henderson, 2003*). Four additional rounds of mode 3 with the resolution limited to 7 Å improved the resolution of the reconstruction to 3.5 Å.

Starting with cycle 6, particles were classified into six classes using 21 rounds of mode 1 refinement. Inspection of the six classes suggested that several represented mixed conformations. The alignment parameters of the class containing the largest number of particles (25%) were therefore used to initialize classification into 15 classes. For this classification, particle images were downsampled by Fourier cropping to a pixel size of 3.28 Å to accelerate processing. 99 rounds of refinement and classification were performed using mode 1 with a resolution limit of 7 Å. To help separate different conformations affecting small subunit, IRES and eEF2, we used a 3D mask that included density belonging to these parts of the structure. This mask was applied in every cycle to the 3D reference structures prior to refinement and classification in 42 additional cycles. The mask was then changed to include only the head of the small subunit, IRES and eEF2, and a final 18 cycles of refinement and classification were run.

We selected six out of the 15 final classes based on clear density present for IRES and eEF2 and continued all further processing with this subset of the data (312,698 particles). The six classes were grouped into three groups based on the rotational state of the small subunit, and each group was further refined and classified using between six and 36 cycles of mode 1 and particles downsampled

to 1.64 Å pixel size. For this classification, FREALIGN's focused mask feature was used to select either the region of IRES PKI (for classes showing intermediate rotation of the small subunit) or a region containing both IRES PKI and eEF2 domain 4 (for classes showing no rotation of the small subunit). This refinement and sub-classification produced eight new classes with more distinct features in the regions selected by the focused masks. These eight classes were used as starting references for a final 33 cycles of refinement and classification using mode 1 and focused mask with the radius of 45 Å covering the vicinity of the ribosomal A site. The first 26 cycles were performed using particles downsampled to 3.28 Å pixel size, followed by two cycles at a pixel size of 1.64 Å, and five cycles at a pixel size of 0.82 Å. The resolution limit for the final cycles was set at 5 Å. The resulting eight reconstructions were used for further analyses, model building and structural refinements, as described below. In parallel, to enhance resolution of the IRES 5′ domain, we performed classification and refinement of the eight classes using a mask with the radius of 50 Å covering the vicinity of the E site and L1 stalk; these maps were used for model building and confirmation of the IRES 5′ domain structure, but not for structure refinements.

Among the resulting eight reconstructions, four reconstructions contained well defined PKI and eEF2 densities (I, II, IV and V) (*Figure 1—figure supplement 1*). PKI was poorly resolved in reconstruction III. Reconstruction VI represents the previously reported 80S•TSV IRES initiation complex in the least rotated conformation (*Koh et al., 2014*). Reconstructions VII and VIII correspond to ribosomes adopting intermediate rotational states, similar to that of Structure III, with weak density in the region of the 5′ domain of the IRES and no density for the PKI domain. To resolve heterogeneity of PKI in reconstruction III, we performed additional sub-classification of all eight classes into two or three classes each. This sub-classification did not result into different structures for the four classes of interest (I, II, IV and V), suggesting a high degree of homogeneity in the masked regions of PKI and eEF2 domain IV. Sub-classification of reconstruction III helped improve the PKI density, resulting in a 4.2 Å reconstruction. All maps were subsequently B-factor-filtered by bfactor.exe (*Lyumkis et al., 2013*), using the B-factors of -50 to -120 Å$^2$, as suggested by bfactor.exe for individual maps, and used for real-space structure refinements. FSC curves (*Figure 1—figure supplement 3*) were calculated by FREALIGN for even and odd particles half-sets. Blocres (*Cardone et al., 2013*) was used to calculate the local resolution of unfiltered and unmasked volumes using a box size 60 pixel, step size of 3 pixels and FSC resolution criterion (threshold 0.143). The output volumes were then colored according to the local resolution of the final reconstructions (*Figure 1—figure supplement 3*) using the Surface Color tool of Chimera (*Pettersen et al., 2004*)

## Model building and refinement

The starting structural models were assembled using the high-resolution crystal structure of *S. cerevisiae* 80S ribosome (*Ben-Shem et al., 2011*), the cryo-EM structure of the 80S•TSV IRES complex (*Koh et al., 2014*) and the crystal structure of the isolated eEF2•sordarin complex (*Jorgensen et al., 2003*). The structure of the diphthamide residue of eEF2 (699) was adopted from PDB: 1ZM4 (*Jorgensen et al., 2005*). Initial domain fitting into the cryo-EM maps was performed using Chimera (*Pettersen et al., 2004*), followed by manual modeling of local regions into the density using Pymol (*DeLano, 2002*) and Coot (*Emsley and Cowtan, 2004*). Parts of several ribosomal proteins were modeled using I-TASSER (*Yang et al., 2015*) and Phyre2 (*Kelley et al., 2015*). The structural models were refined by real-space simulated-annealing refinement using atomic electron scattering factors (*Gonen et al., 2005*), employing RSRef (*Chapman, 1995*; *Korostelev et al., 2002*) as described (*Svidritskiy et al., 2014*). Secondary-structure restrains for ribosomal proteins and base-pairing restraints for RNA molecules were employed, as described (*Laurberg et al., 2008*). The refined structural models closely agree with the corresponding maps, as shown by low real-space R-factors of ~0.2 to 0.27, and they have good stereochemical parameters, characterized by low deviation from ideal bond lengths and angles (*Figure 1—source data 1*). The maps revealed regions, which are differently resolved in Structures I to V. The most prominent difference is in the platform subdomain of the 40S subunit, which is well resolved in Structures I, IV and V but poorly resolved in Structures II and III. The following components of the 40S platform in Structures II and III lacked resolution: proteins eS1, uS11, eS26 and eL41, 18S rRNA nt 892–900, 900–918 and the 3′ end beyond nt 1792. These and other regions of low density were modeled as protein or RNA backbone.

For structural comparisons, the distances and angles were calculated in Pymol and Chimera, respectively. To calculate an angle of the 40S subunit rotation between two 80S structures, the 25S rRNAs were aligned using Pymol, and the angle between 18S rRNAs was measured in Chimera. To calculate an angle of the 40S-head rotation (swivel) between two 80S structures, the 18S rRNAs of the bulk of the 40S body (18S nucleotides excluding nt 1150–1620) were aligned using Pymol, and the angle between the 18S rRNA residues 1150–1620 was measured in Chimera. Figures were prepared in Pymol and Chimera.

## Acknowledgements

We thank Terry Goss Kinzy for providing an eEF2-overexpressing strain of *S cerevisiae*; Rohini Madireddy for assistance with eEF2 purification; Zhiheng Yu and Chuan Hong for assistance with data collection at Janelia Research Campus; Rachel Ruskin, Ruben Diaz-Avalos and Gabriel Demo for assistance with initial data collection and processing; Darryl Conte Jr. for assistance with manuscript preparation; Dmitri Ermolenko, Alexei Korennykh, Harry Noller and members of the Grigorieff and Korostelev laboratories for helpful discussions and comments on the manuscript. This study was supported by NIH Grants R01 GM106105 and GM107465 (to AAK) and R01 GM62580 (to NG).

## Additional information

### Competing interests

NG: Reviewing editor, *eLife*. The other authors declare that no competing interests exist.

### Funding

| Funder | Grant reference number | Author |
| --- | --- | --- |
| National Institutes of Health | GM62580 | Nikolaus Grigorieff |
| Howard Hughes Medical Institute | | Nikolaus Grigorieff |
| National Institutes of Health | GM106105 | Andrei A Korostelev |
| National Institutes of Health | GM107465 | Andrei A Korostelev |

The funders had no role in study design, data collection and interpretation, or the decision to submit the work for publication.

### Author contributions

PDA, Collected and analyzed cryo-EM data, Drafting or revising the article; CSK, Prepared the ribosome•IRES•eEF2 complex, Built and refined structural models, Analysis and interpretation of data, Drafting or revising the article; TG, Assisted with cryo-EM data processing and analyses, Drafting or revising the article; NG, Designed the project, Assisted with cryo-EM data processing and analyses, Drafting or revising the article; AAK, Designed the project, Built and refined structural models, Analysis and interpretation of data, Drafting or revising the article

### Author ORCIDs

Cha San Koh, http://orcid.org/0000-0002-1579-0362
Andrei A Korostelev, http://orcid.org/0000-0003-1588-717X

## Additional files

### Major datasets

The following datasets were generated:

| Author(s) | Year | Dataset title | Dataset URL | Database, license, and accessibility information |
|---|---|---|---|---|
| Priyanka D Abeyrathne, Cha San Koh, Timothy Grant, Nikolaus Grigorieff, Andrei A Korostelev | 2016 | Saccharomyces cerevisiae 80S ribosome bound with elongation factor eEF2-GDP-sordarin and Taura Syndrome Virus IRES, Structure I (fully rotated 40S subunit) | http://www.rcsb.org/pdb/explore/explore.do?structureId=5JUO | Publicly available at the RCSB Protein Data Bank (accession no. 5JUO) |
| Priyanka D Abeyrathne, Cha San Koh, Timothy Grant, Nikolaus Grigorieff, Andrei A Korostelev | 2016 | Saccharomyces cerevisiae 80S ribosome bound to elongation factor eEF2-GDP-sordarin and Taura Syndrome Virus IRES, Structure I | https://www.ebi.ac.uk/pdbe/entry/emdb/EMD-6643 | Publicly available at the Electron Microscopy Data Bank (accession no. EMD-6643) |
| Priyanka D Abeyrathne, Cha San Koh, Timothy Grant, Nikolaus Grigorieff, Andrei A Korostelev | 2016 | Saccharomyces cerevisiae 80S ribosome bound to TSV IRES and eEF2 translation initiation complex, Structure I, Map 2 | https://www.ebi.ac.uk/pdbe/entry/emdb/EMD-6648 | Publicly available at the Electron Microscopy Data Bank (accession no. EMD-6648) |
| Priyanka D Abeyrathne, Cha San Koh, Timothy Grant, Nikolaus Grigorieff, Andrei A Korostelev | 2016 | Saccharomyces cerevisiae 80S ribosome bound to elongation factor eEF2-GDP-sordarin and Taura Syndrome Virus IRES, Structure II (mid-rotated 40S subunit) | http://www.rcsb.org/pdb/explore/explore.do?structureId=5JUP | Publicly available at the RCSB Protein Data Bank (accession no. 5JUP) |
| Priyanka D Abeyrathne, Cha San Koh, Timothy Grant, Nikolaus Grigorieff, Andrei A Korostelev | 2016 | Saccharomyces cerevisiae 80S ribosome bound to elongation factor eEF2-GDP-sordarin and Taura Syndrome Virus IRES, Structure II (mid-rotated 40S subunit) | https://www.ebi.ac.uk/pdbe/entry/emdb/EMD-6644 | Publicly available at the Electron Microscopy Data Bank (accession no. EMD-6644) |
| Priyanka D Abeyrathne, Cha San Koh, Timothy Grant, Nikolaus Grigorieff, Andrei A Korostelev | 2016 | Saccharomyces cerevisiae 80S ribosome bound to TSV IRES and eEF2 translation initiation complex, Structure II, Map 2 | https://www.ebi.ac.uk/pdbe/entry/emdb/EMD-6649 | Publicly available at the Electron Microscopy Data Bank (accession no. EMD-6649) |
| Priyanka D Abeyrathne, Cha San Koh, Timothy Grant, Nikolaus Grigorieff, Andrei A Korostelev | 2016 | Saccharomyces cerevisiae 80S ribosome bound with elongation factor eEF2-GDP-sordarin and Taura Syndrome Virus IRES, Structure III (mid-rotated 40S subunit) | http://www.rcsb.org/pdb/explore/explore.do?structureId=5JUS | Publicly available at the RCSB Protein Data Bank (accession no. 5JUS) |
| Priyanka D Abeyrathne, Cha San Koh, Timothy Grant, Nikolaus Grigorieff, Andrei A Korostelev | 2016 | Saccharomyces cerevisiae 80S ribosome bound to elongation factor eEF2-GDP-sordarin and Taura Syndrome Virus IRES, Structure III (mid-rotated 40S subunit) | https://www.ebi.ac.uk/pdbe/entry/emdb/EMD-6645 | Publicly available at the Electron Microscopy Data Bank (accession no. EMD-6645) |
| Priyanka D Abeyrathne, Cha San Koh, Timothy Grant, Nikolaus Grigorieff, Andrei A Korostelev | 2016 | Saccharomyces cerevisiae 80S ribosome bound to TSV IRES and eEF2 translation initiation complex, Structure III, Map 2 | https://www.ebi.ac.uk/pdbe/entry/emdb/EMD-6650 | Publicly available at the Electron Microscopy Data Bank (accession no. EMD-6650) |
| Priyanka D Abeyrathne, Cha San Koh, Timothy Grant, Nikolaus Grigorieff, Andrei A Korostelev | 2016 | Saccharomyces cerevisiae 80S ribosome bound to TSV IRES and eEF2 translation initiation complex, Structure III, Map 3 | https://www.ebi.ac.uk/pdbe/entry/emdb/EMD-6651 | Publicly available at the Electron Microscopy Data Bank (accession no. EMD-6651) |
| Priyanka D Abeyrathne, Cha San Koh, Timothy Grant, Nikolaus Grigorieff, Andrei A Korostelev | 2016 | Saccharomyces cerevisiae 80S ribosome bound with elongation factor eEF2-GDP-sordarin and Taura Syndrome Virus IRES, structure IV (almost non-rotated 40S subunit) | http://www.rcsb.org/pdb/explore/explore.do?structureId=5JUT | Publicly available at the RCSB Protein Data Bank (accession no. 5JUT) |

| | | | | |
|---|---|---|---|---|
| Priyanka D Abeyrathne, Cha San Koh, Timothy Grant, Nikolaus Grigorieff, Andrei A Korostelev | 2016 | Saccharomyces cerevisiae 80S ribosome bound to elongation factor eEF2-GDP-sordarin and Taura Syndrome Virus IRES, Structure IV (almost non-rotated 40S subunit) | https://www.ebi.ac.uk/pdbe/entry/emdb/EMD-6646 | Publicly available at the Electron Microscopy Data Bank (accession no. EMD-6646) |
| Priyanka D Abeyrathne, Cha San Koh, Timothy Grant, Nikolaus Grigorieff, Andrei A Korostelev | 2016 | Saccharomyces cerevisiae 80S ribosome bound to TSV IRES and eEF2 translation initiation complex, Structure IV, Map 2 | https://www.ebi.ac.uk/pdbe/entry/emdb/EMD-6652 | Publicly available at the Electron Microscopy Data Bank (accession no. EMD-6652) |
| Priyanka D Abeyrathne, Cha San Koh, Timothy Grant, Nikolaus Grigorieff, Andrei A Korostelev | 2016 | Saccharomyces cerevisiae 80S ribosome bound with elongation factor eEF2-GDP-sordarin and Taura Syndrome Virus IRES, Structure V (least rotated 40S subunit) | http://www.rcsb.org/pdb/explore/explore.do?structureId=5JUU | Publicly available at the RCSB Protein Data Bank (accession no. 5JUU) |
| Priyanka D Abeyrathne, Cha San Koh, Timothy Grant, Nikolaus Grigorieff, Andrei A Korostelev | 2016 | Saccharomyces cerevisiae 80S ribosome bound to elongation factor eEF2-GDP-sordarin and Taura Syndrome Virus IRES, Structure V (least rotated 40S subunit) | https://www.ebi.ac.uk/pdbe/entry/emdb/EMD-6647 | Publicly available at the Electron Microscopy Data Bank (accession no. EMD-6647) |
| Priyanka D Abeyrathne, Cha San Koh, Timothy Grant, Nikolaus Grigorieff, Andrei A Korostelev | 2016 | Saccharomyces cerevisiae 80S ribosome bound to TSV IRES and eEF2 translation initiation complex, Structure V, Map 2 | https://www.ebi.ac.uk/pdbe/entry/emdb/EMD-6653 | Publicly available at the Electron Microscopy Data Bank (accession no. EMD-6653) |

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
