## [Decision Letter]

Thank you for submitting your article "Ensemble cryo-EM uncovers inchworm-like translocation of a viral IRES through the ribosome" for consideration by *eLife*. Your article has been favorably evaluated by John Kuriyan (Senior editor) and three reviewers, one of whom, Sriram Subramaniam, is a member of our Board of Reviewing Editors.

The reviewers have discussed the reviews with one another and the Reviewing Editor has drafted this decision to help you prepare a revised submission.

Summary:

Type IV internal ribosome entry sites (IRESs) initiate translation without using any of the canonical eukaryotic translation initiation factors. Thus, they represent the most streamlined mode of eukaryotic translation initiation discovered. They have been studied biochemically and structurally. The current prevailing model is that these IRESs fold into a compact 2-domain structure that bind to the 40S subunit through multiple contacts. Critical interactions occur between two IRES RNA stem-loops and the head of the 40S subunit. This positions one of the IRES domains into the decoding groove. This domain (a pseudoknot, PKI) mimics the tRNA-mRNA anticodon-codon interaction, apparently first docking in the A site. The 60S subunit then joins in a GTP hydrolysis-independent step. This complex is then recognized by elongation factor 2, which catalyzes translocation of the PKI domain into the P site, allowing tRNA delivery to the A-site. Another round of translocation brings this tRNA to the P site. The mechanisms of these IRESs suggests that they can be powerful tools for understand translation in general (a feature the authors of this manuscript exploit).

A rich set of biochemical and functional data have established that different parts of the IRES affect different steps, have shown similarities and differences between different type IV IRESs, and have established some of the key differences between the canonical and IRES-driven initiation mechanisms. In addition, various structures of the IRES alone or bound to the ribosome have been published, using both crystallography and cryo-EM. Until recently, the cryo-EM structures were of low-or mid-resolution. However, when combined with the higher resolution data from crystal structures and the many functional and biochemical studies, the models that resulted have been very informative and have allowed many predictions to be made.

In this manuscript, the authors attack the question of IRES translocation. They present a series of cryo-EM structures of a Taura Syndrome Virus (TSV) IRES bound to 80S and eEF2, using the antibiotic sordarin. The authors interpret the set of structures as showing the trajectory of the mRNA-tRNA-mimicking PKI moving from the A to the P site. Overall, this is an impressive piece of work. It appears technically well done, the description is rich and detailed, and the conclusions are well supported and discussed. As such, it represents an important addition to the IRES field. In many ways, the mechanism that is presented is not surprising; an "inchworm"- like mechanism has been predicted in the literature (although never referred to as such!). However, to see it and to have detailed structures along the trajectory is very important. I will say that as the paper is written, it probably speaks as much to the mechanism of eEF2 and translocation in general as it does to IRES-specific function.

Essential revisions:

1) There are few recent discoveries regarding IRES function that are not mentioned in the manuscript. As these discoveries relate directly to the interactions that the authors visualize and discuss, they should add a bit of discussion or analysis:

• First, some type IV IRESs to can initiate in an alternate reading frame. Do their structures suggest how this might occur? This effect appears to relate to a base adjacent to the codon-anticodon mimic, which they have good density for. References: Au et al. (2015) PNAS 112:E6446-55, Wang et al. (2014) PloS One 9:e103601, Ren et al. (2012) PNAS 109:E630-9, Ren et al. (2014) Nucleic Acids Res. 42:9366-82.

• Recent work implicates the VLR loop/loop 3 in PKI as having a role in eEF2 function: Ruehle et al., (2015) *eLife*), and it has been explored in manuscripts from the Jan lab. This is not mentioned or discussed. Can the authors comment on what this loop is doing and contacting and does it explain this previous work? Also, the Ruehle et al. presents biochemical data in favor of their spontaneous forward and reverse translocation that the authors allude to.

2) The interactions between the highly conserved apical loops of SL4 and 5 make critical interactions with eS5 and eS25. In addition, the IRES makes critical interaction with the L1 stalk. These regions of the type IV IRESs are very highly conserved, but no high-resolution information is known for these interactions. Was the local resolution good enough to say how binding these mysterious interactions are achieved, and perhaps how it relates to ribosome conformation, IRES conformation, etc.?

3) Related to the above, it would be interesting to see some more details of how the IRES changes conformation; not just globally, but internally. Is the resolution sufficient to see this? Any mechanistic insight?

4) By the very nature of this work, in which 5 structures at near atomic resolution are dissected, the figures are quite dense in information content and individual panels are generally quite small. In addition, the paper is quite long because of the high information content. The general reader can of course skip the detailed sections in the middle and read the Discussion, which is very clear. What seems to be missing for a general reader who wishes to dive into the "information forest" is an outline figure at the beginning that shows the ribosome translation cycle with various subunit motions and tRNA movements indicated. This would certainly help those who do not work in the ribosome field.

---

## [Author Response]

1) There are few recent discoveries regarding IRES function that are not mentioned in the manuscript. As these discoveries relate directly to the interactions that the authors visualize and discuss, they should add a bit of discussion or analysis:

First, some type IV IRESs to can initiate in an alternate reading frame. Do their structures suggest how this might occur? This effect appears to relate to a base adjacent to the codon-anticodon mimic, which they have good density for. References: Au et al. (2015) PNAS 112:E6446-55, Wang et al. (2014) PloS One 9:e103601, Ren et al. (2012) PNAS 109:E630-9, Ren et al. (2014) Nucleic Acids Res. 42:9366-82.

We agree that alternative frame selection is an interesting phenomenon andhave added a paragraph to discuss this in “IRES translocation mechanism” (third paragraph). Our structures do not directly suggest how alternate reading frame selection could occur because our data did not reveal a frame-shifted conformation of the IRES. The observation of IRES dynamics in our study indirectly suggests that an alternate (frame-shifted) codon may be transiently placed in the A site following eEF2 release, and this sampling may allow binding of an aminoacyl-tRNA to the off-frame codon.

Recent work implicates the VLR loop/loop 3 in PKI as having a role in eEF2 function: Ruehle et al., (2015) eLife), and it has been explored in manuscripts from the Jan lab. This is not mentioned or discussed. Can the authors comment on what this loop is doing and contacting and does it explain this previous work? Also, the Ruehle et al. presents biochemical data in favor of their spontaneous forward and reverse translocation that the authors allude to.

We have revised our Results section to address this important comment. Loop 3 connects the ASL-like and the mRNA-like regions of the PKI domain. Loop 3 of the post-translocated state (Structure V) is stabilized by interactions with the β-hairpin loop of uS7 and helix 23 of 18S rRNA in the E site, in a manner reminiscent of that for the E-site tRNA in the 80S*2tRNA*mRNA structure. In the pre-translocation states, however, loop 3 is poorly resolved in density maps. This implies conformational flexibility of loop 3, also reported by biochemical studies of unbound IGR IRESs (Jan and Sarnow, 2002; Pfingsten et al., 2007). Our structures therefore suggest that loop 3 contributes to stabilization of the post-translocation IRES, rationalizing the recent detailed biochemical study (Ruehle et al., (2015) *eLife*), which reported that IGR IRES mutated constructs with shortened loop 3 are defective in eEF2-catalyzed translocation.

2) The interactions between the highly conserved apical loops of SL4 and 5 make critical interactions with eS5 and eS25. In addition, the IRES makes critical interaction with the L1 stalk. These regions of the type IV IRESs are very highly conserved, but no high-resolution information is known for these interactions. Was the local resolution good enough to say how binding these mysterious interactions are achieved, and perhaps how it relates to ribosome conformation, IRES conformation, etc.?

We find that the phosphate backbone of SL4 and 5 interact with the positively charged and aromatic residues of eS5 (uS7) and eS25. We have added a description of these interactions in the main text and also Supplementary Figures (Figure 3—figure supplement 2, Figure 3—figure supplement 4) to demonstrate the interactions. In addition, we find that interactions of SL4 and SL5 with the small subunit are somewhat similar to those of the L1 stalk with the small subunit of the hybrid-state 80S*tRNA structure. We have added Figure 3—figure supplement 3 to illustrate this similarity. The interactions between the IRES and the L1 stalk are less well resolved – although the density is strong, the resolution is insufficient to interpret the interactions unambiguously. We therefore refrain from detailed interpretation of L1 stalk interactions.

3) Related to the above, it would be interesting to see some more details of how the IRES changes conformation; not just globally, but internally. Is the resolution sufficient to see this? Any mechanistic insight?

We now provide a more extensive discussion of IRES local interactions and conformational changes, supplemented by additional illustrations. We report the rearrangements of stem loop 3 (conserved in TSV-like IRESs of group 2: *Aparavirus*), which resembles a tRNA elbow, as we reported previously. Our current structures indicate that SL3 undergoes rearrangements similar to those of the translocating A-site tRNA (Figure 1—figure supplement 6). In addition, we demonstrate local rearrangements of the “bridge” between the 5’ domain and PKI domain (Figure 3—figure supplement 7). This region interacts with protein uL5 in the two most compact IRES conformations (III and IV), but not in other states. This reveals the stabilizing role of protein uL5 at the intermediate stages of IRES translocation.

4) By the very nature of this work, in which 5 structures at near atomic resolution are dissected, the figures are quite dense in information content and individual panels are generally quite small. In addition, the paper is quite long because of the high information content. The general reader can of course skip the detailed sections in the middle and read the Discussion, which is very clear. What seems to be missing for a general reader who wishes to dive into the "information forest" is an outline figure at the beginning that shows the ribosome translation cycle with various subunit motions and tRNA movements indicated. This would certainly help those who do not work in the ribosome field.

We have reorganized the panels in most figures to make the figures less dense and increase the sizes of individual panels. We agree that a figure showing ribosome-2tRNA-mRNA and summarizing conformational differences between structures representing various translocation states would be helpful. We now include a supplementary figure, showing ribosome-2tRNAs-mRNA structures (Figure 1—figure supplement 1), which we refer to in the manuscript. We also include the views of tRNA-bound structures in the supplementary figure showing interactions of the A-site finger with the tRNAs and the IRES (Figure 3—figure supplement 6).